# Ionic mechanisms underlying history-dependence of conduction delay in an unmyelinated axon

**Yang Zhang[1], Dirk Bucher[2], Farzan Nadim[1,2]\***

[1]Department of Mathematical Sciences, New Jersey Institute of Technology, Newark, United States; [2]Federated Department of Biological Sciences, NJIT and Rutgers University, Newark, United States

**Abstract** Axonal conduction velocity can change substantially during ongoing activity, thus modifying spike interval structures and, potentially, temporal coding. We used a biophysical model to unmask mechanisms underlying the history-dependence of conduction. The model replicates activity in the unmyelinated axon of the crustacean stomatogastric pyloric dilator neuron. At the timescale of a single burst, conduction delay has a non-monotonic relationship with instantaneous frequency, which depends on the gating rates of the fast voltage-gated $Na^+$ current. At the slower timescale of minutes, the mean value and variability of conduction delay increase. These effects are because of hyperpolarization of the baseline membrane potential by the $Na^+/K^+$ pump, balanced by an h-current, both of which affect the gating of the $Na^+$ current. We explore the mechanisms of history-dependence of conduction delay in axons and develop an empirical equation that accurately predicts this history-dependence, both in the model and in experimental measurements.

\*For correspondence: farzan@njit.edu

**Competing interests:** The authors declare that no competing interests exist.

## Introduction

Recent years have seen a growing appreciation of the role of axons in neural signaling. Rather than serving solely as faithful transmission lines of the neural code, axons shape neuron output through activity- and neuromodulator-dependent changes in propagation fidelity (*Debanne, 2004*; *Alle and Geiger, 2008*; *Kress and Mennerick, 2009*; *Bucher and Goaillard, 2011*; *Debanne et al., 2011*; *Sasaki, 2013*; *Bucher, 2015*). Repetitive activity alters axonal membrane excitability and conduction velocity, which can substantially change the temporal pattern of spikes during propagation from proximal initiation sites to distal presynaptic sites. Apart from the more extreme cases of spike failures or ectopic spike initiation, changes in conduction velocity across consecutive spikes result in varying conduction delays, and therefore in altered spike intervals. Temporal coding schemes, in which information is contained in spike intervals, latencies, or firing phase (*Panzeri et al., 2010*), may be exquisitely sensitive to such changes.

Many of the activity-dependent effects on spike propagation are owed to the fact that membrane excitability in most axons is substantially more complex than often credited. In addition to the basic ionic mechanisms necessary for spike propagation, i.e., fast sodium and delayed rectifier potassium conductances (*Hodgkin and Huxley, 1952a*, *1952b*), axons can have a large variety of voltage-gated ion channels and ion pumps (*Krishnan et al., 2009*; *Bucher and Goaillard, 2011*; *Debanne et al., 2011*). As the activation and gating of these currents can be associated with a wide range of time constants, conduction velocity can change depending on the history of activity at timescales far exceeding refractory effects occurring in the millisecond range (*George, 1977*; *Raymond, 1979*; *Weidner et al., 2002*; *Bucher and Goaillard, 2011*; *Ballo et al., 2012*). However, the ionic mechanisms underlying history-dependence of spike propagation are not well understood, and

the common approach to test excitability changes with simple paired conditioning and test pulses (*Bostock et al., 1998*; *Krishnan et al., 2009*; *Bucher and Goaillard, 2011*) is insufficient to capture slow dynamics and transformation of complex temporal patterns (*Weidner et al., 2002*; *Ballo et al., 2012*).

Conduction velocity in unmyelinated axons depends on membrane and cytosol properties, and axon diameter (*Hodgkin and Huxley, 1952a*; *Hodgkin, 1954*; *Del Castillo and Moore, 1959*; *Waxman, 1975*; *Bucher and Goaillard, 2011*). Total conductance change during the spike (*Matsumoto and Tasaki, 1977*; *Tasaki and Matsumoto, 2002*; *Tasaki, 2004*), as well as sodium channel gating (*Muratov, 2000*), can be used to accurately predict conduction velocity of isolated spikes. It is unclear, however, how well those descriptions apply to complex excitability changes during highly repetitive activity. Additionally, theoretical investigations of the history-dependence of conduction delay have considered only minimal sets of ionic currents (*Miller and Rinzel, 1981*; *Kepler and Marder, 1993*; *Moradmand and Goldfinger, 1995*; *Faisal and Laughlin, 2007*).

Here we study the ionic mechanisms underlying history-dependence of conduction delay at different timescales, using a biophysical axon model with relatively complex excitability. The model was inspired by experimental findings from the pyloric dilator (PD) motor axon of the lobster *Homarus americanus* stomatogastric nervous system. This several-centimeters long axon shows large changes in conduction delay during repetitive stimulation (*Ballo and Bucher, 2009*; *Ballo et al., 2012*). Delays depend on spike intervals, but this dependence changes substantially over the course of several minutes of stimulation. Furthermore, axon excitability critically depends on modulation by dopamine, which drastically alters temporal fidelity (*Ballo et al., 2010*, *2012*).

Our model replicates this behavior well. We find that delay changes are mostly because of the effect of membrane potential changes on a subset of variables describing sodium channel gating. Other currents contribute via their effect on the membrane potential. Equations developed to predict conduction velocity of isolated spikes fail to capture activity-dependent changes. We therefore develop empirical equations that allow good predictions of delay changes from either sodium channel gating variables or voltage trajectories. The latter also perform well on experimental data.

## Results

### The history-dependence of conduction delay in the model axon is similar to experimental observations in the PD axon

To examine dependence of a discrete process on the history of prior activity it is common to use *Poisson* train stimuli (*Krausz and Friesen, 1977*; *Sen et al., 1996*; *Stern et al., 2009*), which by definition include a large range of inherent frequencies. *Poisson* trains have been used before in Hodgkin-Huxley model axons to characterize spike delay and velocity (*George, 1977*; *Moradmand and Goldfinger, 1995*), but not commonly in model axons with more complex excitability. We therefore examined the history-dependence of conduction delay in the PD neuron model axon by stimulating one end of the axon and measuring conduction delay between two positions along its length (*Figure 1*).

Experimental data from *Poisson* stimulation of the PD axon show a slow hyperpolarization that is accompanied by overall slowing of propagation (*Ballo et al., 2012*). History-dependence of delay is manifest in several ways. At the slow timescale (STS) of hyperpolarization, both the mean value and variability of conduction delay increase over a timescale of minutes following the onset of the stimuli (*Figure 2A*).

At a fast timescale (FTS), a nonlinear and non-monotonic relationship between delay and the instantaneous stimulus frequency ($F_{inst}$) is observed: conduction delay of the PD axon has a minimum value for $F_{inst}$ around 40 Hz but higher values for lower or higher $F_{inst}$ (*Figure 2B*). Note that the FTS effect compares the conduction delay of each spike only to that of the previous one. The FTS effect, that is, the nonlinear and non-monotonic frequency-dependence, changes with the STS effect, as can be seen in *Figure 2B* in the changing frequency-dependence at different times of the stimulation. These effects are strongest during pharmacological block of $I_h$, presumably because the axon hyperpolarizes more in the absence of inward rectification (*Ballo et al., 2012*, see below). The data shown in *Figure 2A and B* were obtained under these conditions. To match these conditions, the simulations shown in *Figure 2C and D* were performed without $I_h$ in the model ($\bar{g}_h$=0), but with $I_{pump}$

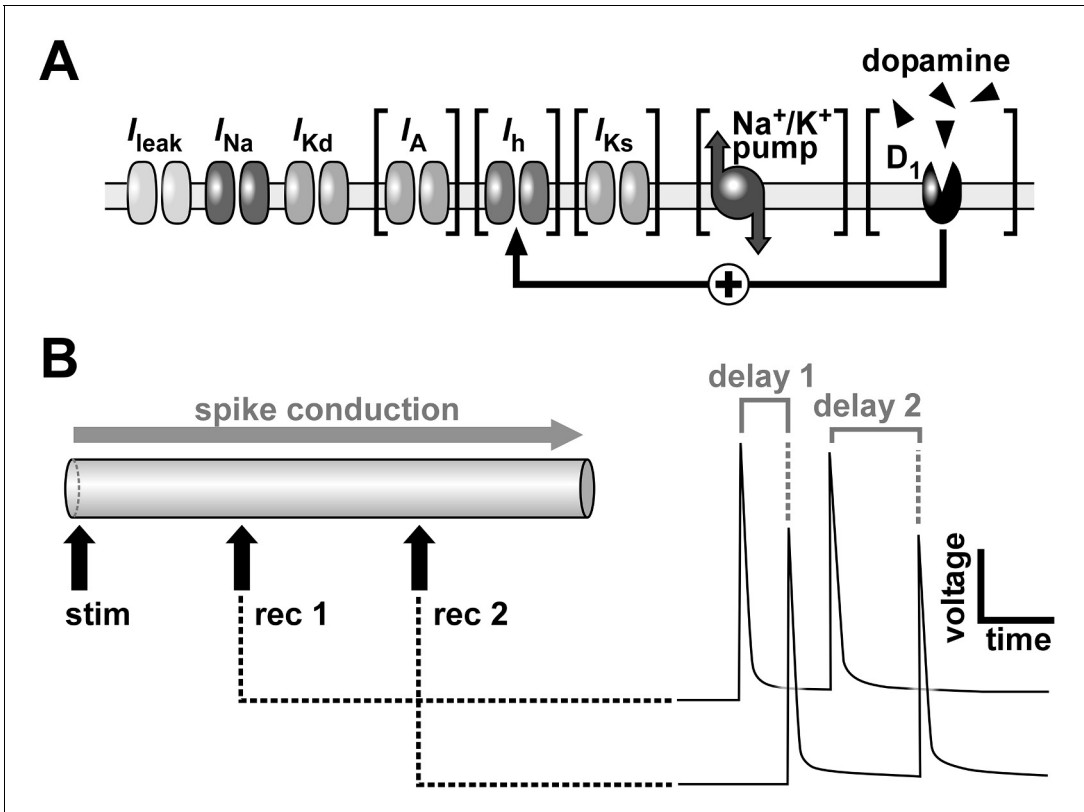

**Figure 1.** Schematic of the model axon and conduction delay. (**A**) Schematic of ion channels, the Na$^+$/K$^+$ pump and dopamine receptors in the model axon. Brackets indicate components that were omitted in some simulations. (**B**) The model axon was stimulated at one end and the conduction delay was measured between two positions at 0.3 and 0.7 times the length of the axon. Right panel shows conduction delays between the recording sites for two consecutive action potentials.

and $I_A$ in addition to $I_{Leak}$, $I_{Na}$ and $I_{Kd}$. We note that both the STS and the FTS effects were present with or without $I_h$ in the model and, later on, we will show how $I_h$ influences these effects in the model and experiments. The model captured, qualitatively and quantitatively, both the STS and the FTS history-dependence of conduction delay observed in the experiments (*Figure 2C and D*). To compare the experimental and model STS effect, we calculated the mean ($D_{mean}$) and coefficient of variation (CV-D) of conduction delay in 20 s time bins. The model provided an excellent match to the changes in $D_{mean}$ and the increase in variability over the 5 min stimulation period (*Figure 2E and F*).

To test to which degree the STS and FTS history-dependence are the result of more complex membrane excitability, we applied the same *Poisson* stimulation to an axon modeled using the classical Hodgkin-Huxley equations (not including $I_A$, $I_h$, and $I_{pump}$). The results of this simulation indicated that the Hodgkin-Huxley model axon showed no slow history-dependence (*Figure 2G*). It did, however, show a weak FTS history-dependence for $F_{inst}$ values larger than 40 Hz (*Figure 2H*). This effect was qualitatively similar to the FTS effect seen in the PD model axon (*Figure 2D*).

### The STS effect is determined by the activity level of the Na$^+$/K$^+$ pump

The STS effect occurred over a timescale of minutes and should therefore be related to a slow activity-dependent process of the PD axon. In many axons, slowing of conduction is thought to be because of hyperpolarization caused by the Na$^+$/K$^+$ pump (*Bucher and Goaillard, 2011*), and our model did not produce any STS effect without the pump (not shown).

The activity level of $I_{pump}$ is determined by the fast sodium current (see Materials and methods). Therefore, both the increase of $I_{pump}$ with time, and the steady state level it reached, depended on the mean rate of stimulation (*Figure 3A*). *Figure 3B and C* show that both $D_{mean}$ and CV-D also

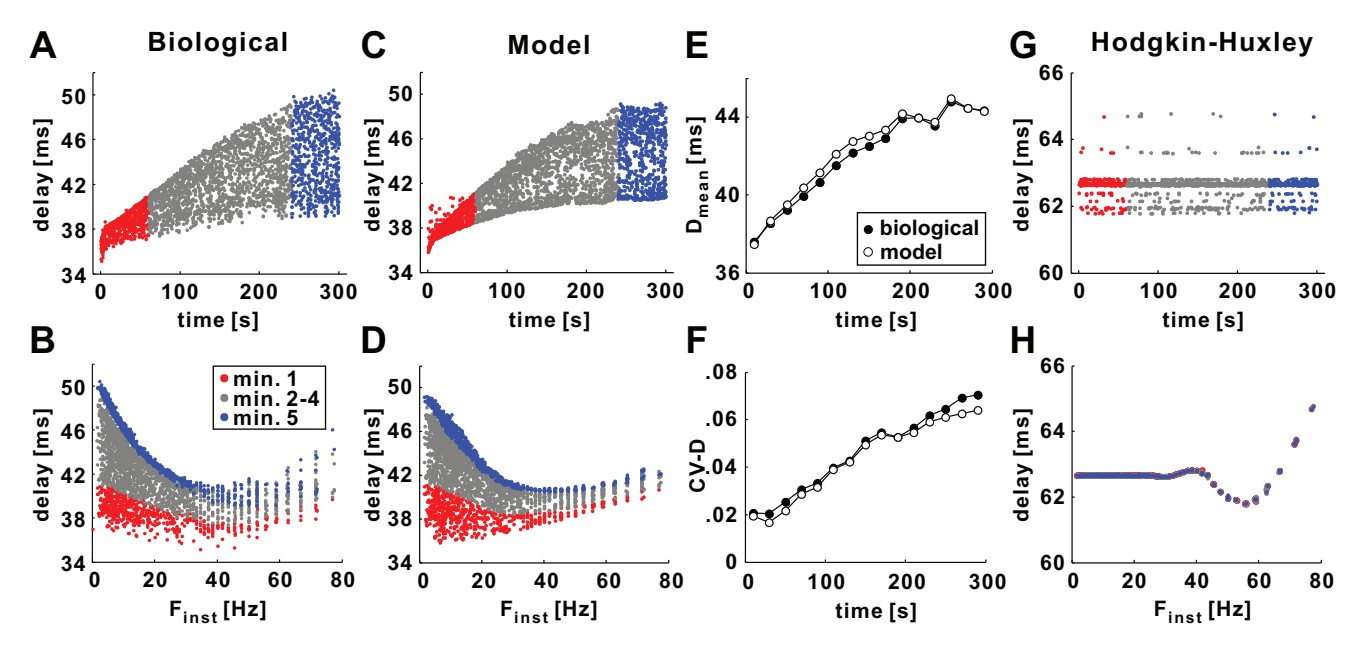

**Figure 2.** History-dependence of conduction delay in the model axon (with $g_h = 0$). (A) Delay shown as a function of stimulus time for a 10 Hz (mean rate), 5 min *Poisson* stimulation of the axon of the biological PD neuron. Red and blue, respectively, mark delay values for the first and fifth minute of stimulation. (B) The data in panel *A*, plotted as a function of the instantaneous stimulus frequency ($F_{inst}$). Colors as in panel *A*. (C–D) The same *Poisson* stimulation as in panel *A*, applied to the model axon produces results that are quantitatively similar to the experimental data. (E–F) The mean ($D_{mean}$) and the coefficient variation (CV-D) of the data in panels *A* and *C*, measured in 20 s bins. (G–H) The same *Poisson* stimulation of panels *C-D*, applied to the standard Hodgkin-Huxley model axon.

increased depending on the stimulation rate, and with time courses very similar to $I_{pump}$. With increased stimulation, $I_{pump}$ produced a hyperpolarization of the baseline membrane potential (*Figure 3D*). When $I_{pump}$ was modeled without dynamics and instead set to different constant values during *Poisson* stimulation, the baseline membrane potential (trough voltage, $V_T$) showed a linear dependence on $I_{pump}$, while the peak voltage of spikes ($V_P$) was fairly unaffected (*Figure 3E*). Both $D_{mean}$ and CV-D appeared to be determined by the level of $I_{pump}$, as both showed a positive and linear dependence (*Figure 3F*). We therefore concluded that the dominant cause of the STS effect is the hyperpolarization mediated by $I_{pump}$.

## A slow K$^+$ current fails to produce the STS effect

Alternatively, slow hyperpolarization could be because of cumulative activation of slow K$^+$ currents. We therefore tested if a slow conductance can produce a similar STS effect as $I_{pump}$. To address this possibility, we replaced the Na$^+$/K$^+$ pump with a slowly-activating potassium current ($I_{Ks}$) which was tuned to produce a slow increase of conductance on a timescale of several minutes. As before, $\bar{g}_h$ was set to zero.

The voltage activity of the model axon with $I_{Ks}$ showed a slow decrease in $V_P$ but little change in $V_T$ (*Figure 4A*). This was in contrast to the effect of $I_{pump}$, which had a strong effect in hyperpolarizing $V_T$ but little effect on $V_P$ (*Figure 3D and E*). Although $g_{Ks}$ increased with a similar time course as $I_{pump}$ in the model version described in *Figure 3*, the behavior of $I_{Ks}$ was quite different. Unlike the slow continuous increase of $I_{pump}$ (*Figure 3A*), $I_{Ks}$ increased with each action potential and was reset back to almost 0 between action potentials (*Figure 4B*, expanded in 4C) and only the peak value of $I_{Ks}$ increased with a similar time course as $g_{Ks}$. As a result, $I_{Ks}$ was only effective during each spike (reducing $V_P$) and had no cumulative effect in between spikes, and therefore could not change the baseline membrane potential. The lack of effect of $I_{Ks}$ on $V_T$ is because of the small driving force of this current during the inter-spike intervals. Interestingly, $D_{mean}$ increased with a similar time course

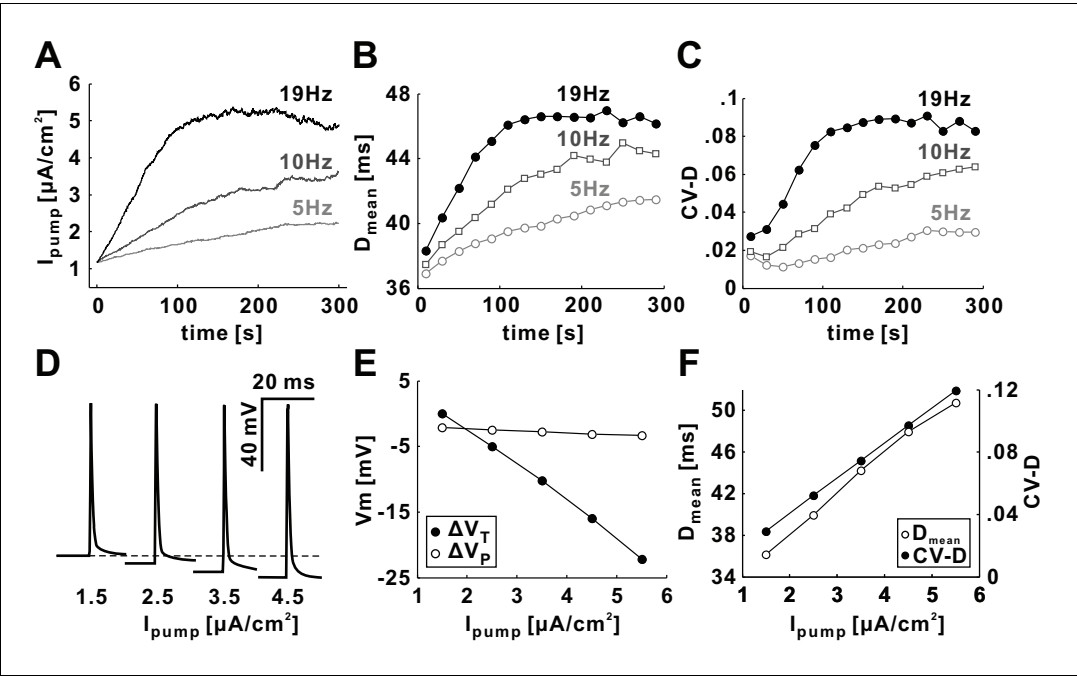

**Figure 3.** $D_{mean}$ and CV-D strongly depend on the activity level of $I_{pump}$ at all *Poisson* stimulation frequencies. **(A)** The value of $I_{pump}$ for *Poisson* stimulations with different mean rates (5, 10 and 19 Hz). **(B–C)** $D_{mean}$ and CV-D increase with time as well as with the mean rate of *Poisson* stimulation. **(D)** The baseline membrane potential is hyperpolarized as the level of $I_{pump}$ is increased. **(E)** The trough voltage ($V_T$) linearly decreases with $I_{pump}$, but the peak voltage ($V_P$) does not change much when $I_{pump}$ is increased. **(F)** $D_{mean}$ and CV-D linearly increase with the level of $I_{pump}$ (set to a constant value in each simulation run).

as seen in the decrease of $V_P$ (*Figure 4D and E*), and this increase was only slightly smaller than in the model including $I_{pump}$. However, the CV-D stayed relatively constant throughout *Poisson* stimulation (*Figure 4D and F*). These results suggest that the STS effect on the variability of delay strongly depends on the baseline membrane potential, while increase in mean delay can be because of baseline hyperpolarization or decreasing peak voltage.

We conclude that the STS effect on variability of delay in the case of the PD axon could not easily be captured with $I_{Ks}$, possibly because the minimal driving force between spikes did not allow for a substantial cumulative hyperpolarization.

## The dependence of conduction delay on $I_h$

In the PD axon, inward rectification by $I_h$ plays an important role during repetitive firing. It is modulated by dopamine (DA), which increases $g_h$ 2–3 fold in the range of normal resting membrane potentials (*Ballo et al., 2010*). DA prevents hyperpolarization and drastically reduces history-dependence of conduction delay, while pharmacological block of $I_h$ increases both hyperpolarization and history-dependence (*Ballo et al., 2012*). Here, we wanted to examine whether the experimentally-observed effects of DA and CsCl on $I_h$ levels are sufficient to explain their effects on the history-dependence of conduction delay.

We approximated the effects of DA enhancement or pharmacological block of $I_h$ by adjusting the model parameters governing this current (see Materials and methods and *Tables 1* and *2*), and we will refer to these respective model parameter sets as DA or $I_h$ block. We examined how the STS and FTS effects were influenced by DA or $I_h$ block. Such a comparison is shown in *Figure 5* for the *Poisson* stimulation with a mean rate of 10 Hz. Blocking $I_h$ increased both $D_{mean}$ and CV-D in the model, an effect that qualitatively matched the experimental application of CsCl (*Figure 5A,B*). Consistent with experimental data, DA had the opposite influence on the STS effect.

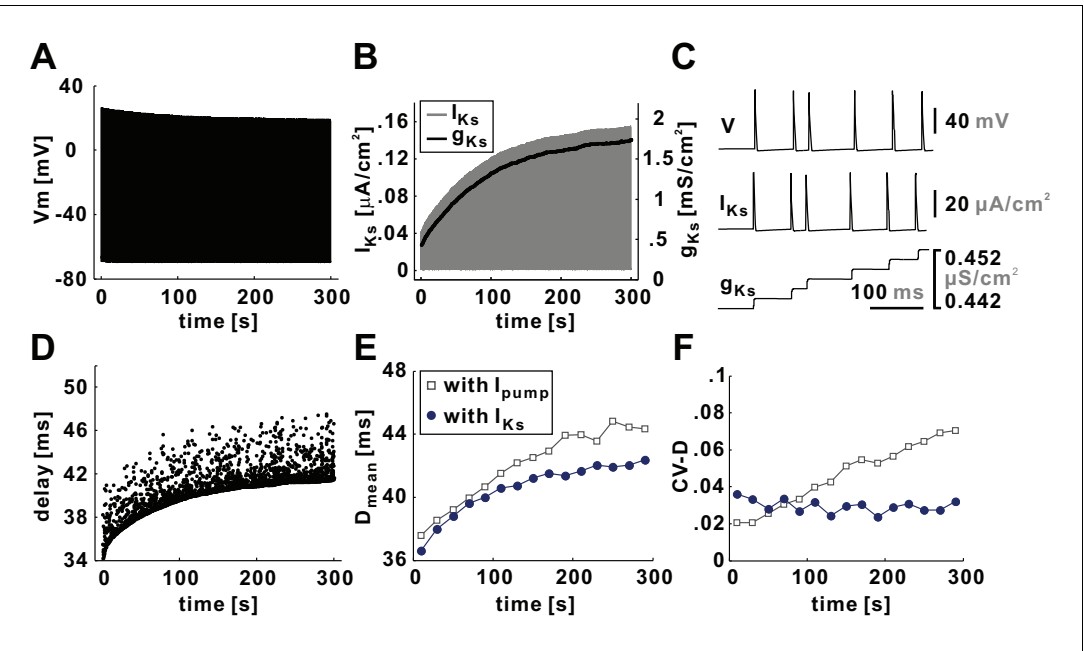

**Figure 4.** A slow potassium current ($I_{Ks}$) fails to capture the slow timescale effects on conduction delay. (**A**) Voltage trace of the model axon, with $I_{Ks}$ substituted for $I_{pump}$, for a 5 min, 10 Hz *Poisson* stimulation. (**B**) Both $g_{Ks}$ and the peak values of $I_{Ks}$ increase with time during the stimulation process. (**C**) A small time-window shows that, although $g_{Ks}$ accumulates with time, $I_{Ks}$ vanishes after each action potential because of the drop in the driving force. (**D**) Conduction delay of the model axon in response to a 5 min, 10 Hz *Poisson* stimulation. (**E**) $D_{mean}$ increases with time but does not quite match the results of the model axon with $I_{pump}$. (**F**) CV-D does not increase with time as it does in the model axon with $I_{pump}$.

To quantify the influence of $I_h$ on the FTS effect, we focused on the data in the fifth minute of the *Poisson* stimulation (***Figure 5C***). We fit the nonlinear relationship between delay and $F_{inst}$ with a quadratic function and determined $\kappa_{min}$, the curvature at the frequency at which delay was smallest, as a

**Table 1.** Model parameters.

| Ionic Curr | $\bar{g}_x$ [mS/cm²] | $E_x$ [mV] |
|---|---|---|
| $I_{Na}$ | 14 | Dynamic |
| $I_{Kd}$ | 3 | −70 |
| $I_{Leak}$ | 0.125 | −65 |
| $I_A$ | 5 | −70 |
| $I_h$ (ctrl) | 0.05 | −32 |
| $I_h$ (DA) | 0.1 | −25 |
| $I_{Ks}$ | 3 | −70 |
| *Other* | *Value* | |
| $I_{max,pump}$ | 2 mA/cm² | |
| $[Na^+]_{1/2}$ | 78 mM | |
| $[Na^+]_S$ | 2 mM | |
| $\alpha$ | 7.4 | |
| *Vol* | 7850 µm³ | |
| $F$ | 96485 C/mol | |

**Table 2.** Parameters of ionic current steady-state activation and inactivation and their time constants.

| Ionic Curr | m, h | $x_\infty$ | $\tau_x$ [ms] |
|---|---|---|---|
| $I_{Na}$ | $m^3$ | $\frac{1}{1+exp(-(V+48)/8.5)}$ | $\frac{0.132}{cosh((V+27)/7.5)} + \frac{0.003}{1+exp(-(V+27)/5)}$ |
|  | $h$ | $\frac{1}{1+exp((V+47)/6)}$ | $\frac{10}{cosh((V+42)/15)}$ |
| $I_{Kd}$ | $m^4$ | $\frac{1}{1+exp(-(V+47)/10)}$ | $\frac{50}{cosh((V+73)/15)}$ |
| $I_A$ | $m^3$ | $\frac{1}{1+exp(-(V+63)/15)}$ | $18 + \frac{58}{1+exp((V+61)/20)}$ |
|  | $h$ | $\frac{1}{1+exp((V+80)/8)}$ | $50$ |
| $I_h$ (ctrl) | $m$ | $\frac{1}{1+exp((V+80)/5.5)}$ | $3700$ |
| $I_h$ (DA) | $m$ | $\frac{1}{1+exp((V+75)/12.5)}$ | $3800$ |
| $I_{Ks}$ | $m$ | $\frac{1}{1+exp(-(V+47)/10)}$ | $\frac{250000}{cosh((V+73)/15)}$ |

measure of nonlinearity. Blocking $I_h$ increased the nonlinearity in the $F_{inst}$-delay relationship (**Figure 5C2**; control: $\kappa_{min}$=0.0059; $\bar{g}_h$ = 0: $\kappa_{min}$=0.0091), whereas DA decreased the nonlinearity (2 x control $\bar{g}_h$: $\kappa_{min}$=0.0032). These effects mimic the influence of CsCl and DA on the FTS effect in experiments (**Figure 5C1**; control: $\kappa_{min}$=0.0051; CsCl: $\kappa_{min}$=0.01; DA: $\kappa_{min}$=0.001).

Although the biophysical model qualitatively captured the STS and FTS effects of $I_h$ changes seen in biological experiments, there were small quantitative differences between the model and the experiments. These were most notable for the DA effect when the simulation $D_{mean}$ values were

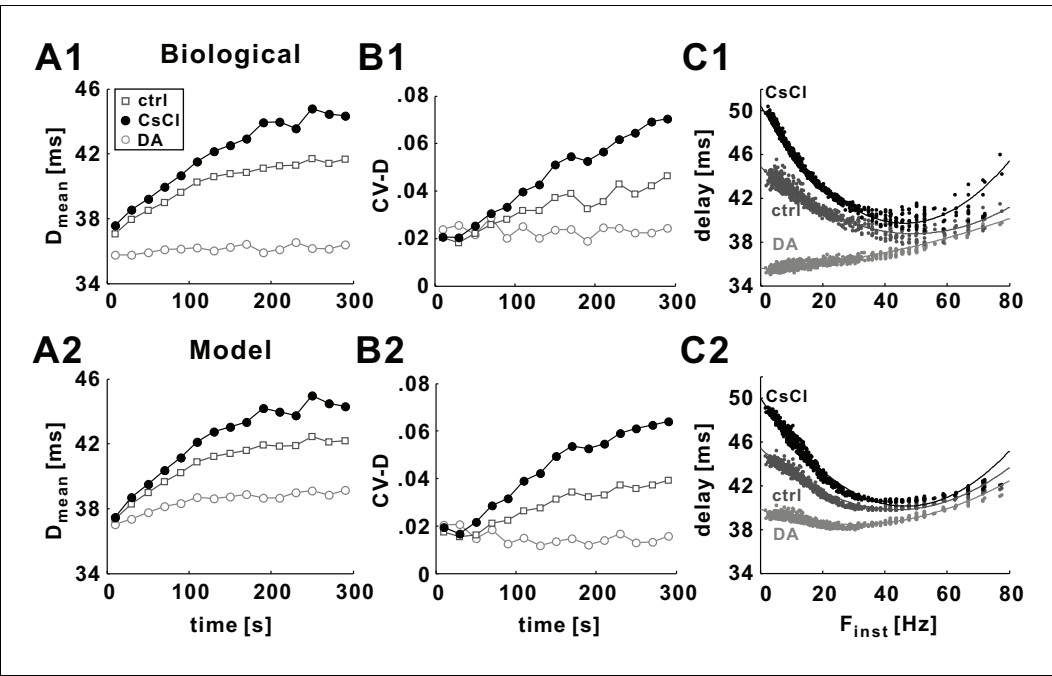

**Figure 5.** Changing the level of $I_h$ in the model mimics the experimental effects of CsCl and DA. (**A1–B1**) At the slow timescale, DA results in an increase of $g_h$, which in turn causes $D_{mean}$ and CV-D stay to remain relatively constant in response to a 5 min 10 Hz *Poisson* stimulation. In contrast, blocking $I_h$ with CsCl results in an increase in $D_{mean}$ and CV-D (**Ballo et al., 2012**). (**A2–B2**) This effect is mimicked by changing $g_h$ levels and the activation curve of $I_h$ in the model. (**C1**) Changes in $g_h$ levels by DA or CsCl result in changes in the minimum delay ($D_{min}$) and curvature ($\kappa_{min}$) values in the delay vs. $F_{inst}$ plots (for the 5[th] minute of the *Poisson* stimulation). (**C2**) These effects are mimicked by the models with different activity level of $I_h$ (see Materials and methods and **Tables 1** and **2** for details).

larger than the experimental values (*Figure 5A1–A2*), and there was a small increase in D$_{mean}$ that was not observed in the biological data. Additionally, the nonlinearity of the FTS effect was reduced but not removed as in the experiments (*Figure 5C1–C2*).

## The FTS effects can be predicted by paired-pulse stimulation protocols

Activity-dependent changes in axon excitability and conduction velocity are often gauged with paired-pulse stimulations (*Bucher and Goaillard, 2011*), particularly for diagnostic purposes in the context of peripheral neuropathies (*Bostock and Rothwell, 1997*; *Bostock et al., 1998*; *Burke et al., 2001*; *Krishnan et al., 2009*). Such stimulation protocols consist of a single conditioning pulse followed by a test pulse delivered at different intervals. The change in spike threshold, conduction velocity, or conduction delay between conditioning and test pulses can then be analyzed as a function of inter-stimulus interval (ISI). Typically, excitability changes as a function of ISI in a sequence of relative refractory, supernormal, and sometimes subnormal intervals, collectively referred to as the 'recovery cycle'. Recovery cycle measurements gauge excitability changes occurring at a similar timescale as the FTS effect seen in *Poisson* stimulations. We predicted that employing paired-pulse stimulations to test conduction velocity changes is equivalent to analyzing the FTS effect on conduction delay at the beginning of the *Poisson* stimulation.

Instead of conditioning trains of different duration or frequency, we used simple paired-pulse stimulations and mimicked different excitability states of the axon by setting $I_{pump}$ to different constant values. These values were obtained from the 10 Hz *Poisson* stimulation results described in *Figure 3A*, the relatively low mean value during minute 1 ($I_{pump,LO}$), and the relatively high mean

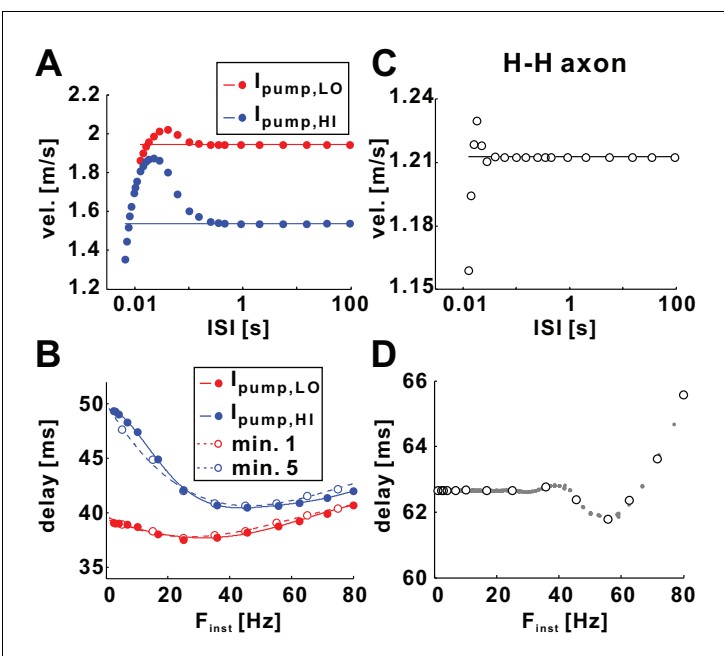

**Figure 6.** The fast timescale effect can be predicted by paired-pulse stimulations. (**A**) Conduction velocity of conditioning (horizontal line) and test (filled circles) pulses plotted as a function of interspike interval (ISI). The graph shows the refractory and supernormal phases, where the velocity of the test pulse is, respectively, less or more than that of the conditioning pulse. Simulations were done with two constant pump currents: $I_{pump,LO}$ and $I_{pump,HI}$, respectively equal to the mean value of $I_{pump}$ during minute 1 and minute 5 of the 10 Hz *Poisson* stimulation (*Figure 2*). Horizontal lines show the velocity of the conditioning spikes. (**B**) Data of panel *A* plotted as conduction delay vs. F$_{inst}$ (=1/ISI), together with the *Poisson* stimulation results of minutes 1 and 5 (from *Figure 2D*). Solid curves are polynomial fits for the $I_{pump,LO}$ and $I_{pump,HI}$ datasets. (**C**) Paired-pulse stimulations of the Hodgkin-Huxley model axon show similar refractory and supernormal phases. (**D**) The data from the panel *C* compared with the *Poisson* stimulation data (from *Figure 2H*) of the Hodgkin-Huxley model axon show a perfect match.

value during minute 5 ($I_{pump,HI}$). *Figure 6A* shows recovery cycle measurements for both levels of $I_{pump}$. Consistent with the STS effects of $I_{pump}$, the conduction velocities of the conditioning (line) and test spikes were higher with $I_{pump,LO}$ than with $I_{pump,HI}$. For both levels of $I_{pump}$, test spikes showed reduced velocity at small ISIs compared to the conditioning spike (relative refractory period), and increased velocities at larger ISIs (supernormal period), before converging back to the velocity of the conditioning pulse at even larger ISIs. However, note that with different levels of $I_{pump}$, the peak conduction velocity of the test pulse corresponded to different ISIs and that the difference between the velocity of conditioning pulse and the peak velocity of the test pulse was larger with $I_{pump,HI}$.

In order to compare the paired-pulse data with the FTS effect seen in the delay vs. $F_{inst}$ relationship of the *Poisson* stimulation (*Figure 2D*), we plotted the data from *Figure 6A* as delay vs. $F_{inst}$ and fit these data with a cubic polynomial function. *Figure 6B* shows plots of these functions together with the data for the first and the fifth minute of the *Poisson* stimulation (the latter in 10 Hz bins). This comparison showed that the fits of the paired-pulse data provided very good predictions of the FTS effect seen in the *Poisson* stimulation ($R^2 = 0.939$ for minute 1 and $I_{pump,LO}$; $R^2 = 0.937$ for minute 5 and $I_{pump,HI}$). We can therefore conclude that at the fast timescale, delay is primarily determined by the ISI, that is, the timing of the last preceding spike. The STS effect influences the dependence of velocity or delay on ISI or $F_{inst}$, but this effect is slow enough that the FTS effect exposed by *Poisson* stimulation can still be accurately predicted by simple paired-pulse or train-pulse stimulation methods.

We also compared the effect of paired-pulse and *Poisson* stimulations in the Hodgkin-Huxley model axon. As shown in *Figure 2*, this model exhibits no STS effect. We therefore predicted that recovery cycle measurements should perfectly match data from *Poisson* stimulations, even when repetitive activity is used as a conditioning paradigm. To test this, we used a single pulse as conditioning stimulus. The recovery cycle measurement in *Figure 6C* shows that relative refractory and supernormal periods were restricted to relatively small ISIs. *Figure 6D* shows that when plotted as delay vs. $F_{inst}$, results from paired-pulse stimulations are a near perfect match to results from *Poisson* stimulations.

## The FTS effects of the PD axon during ongoing bursting can be predicted by paired-pulse stimulation

The PD neuron is a member of the pacemaker group of the pyloric network and its natural ongoing activity consists of bursts with a cycle frequency of ~1 Hz and a parabolic interval structure within each burst (*Szucs et al., 2003*; *Ballo and Bucher, 2009*; *Ballo et al., 2012*). During ongoing bursting activity, different spikes of each burst have different conduction delays and there is a highly nonlinear relationship between the conduction delay and the spike number in the burst (*Ballo and Bucher, 2009*; *Ballo et al., 2012*). We used the natural activity pattern in our model to address two questions. First, we examined whether the natural activity changes the STS and FTS effects or whether these effects remain the same as during *Poisson* stimulations. Second, we asked if the FTS effect for the natural bursting activity could be as accurately predicted from paired-pulse stimulations as for *Poisson* stimulations.

*Figure 7A* shows traces of the model axon from the start and end of a 300 s stimulation with a burst frequency of 1 Hz. Action potentials were produced by stimulating the model with brief current pulses (see Materials and methods) and it captured the dynamics of voltage trajectories described in the experimental data (*Ballo and Bucher, 2009*). In particular, over the course of the entire stimulation, the baseline membrane potential between bursts slowly hyperpolarized by several millivolts. Within each burst, both the peak and the trough voltages changed. Peak voltages decreased with increasing $F_{inst}$ towards the middle of bursts and increased again with decreasing $F_{inst}$ towards the end, because of inactivation of $I_{Na}$. Trough voltages were more depolarized in the middle of bursts, because summation resulting from relatively slow spike repolarization is largest at high $F_{inst}$.

*Figure 7B1* shows PD axon conduction delays over the course of the burst duration for all 300 bursts in the 5 min stimulation, measured in one experiment. As in *Poisson* stimulations, $D_{mean}$ and CV-D increased during the entire stimulation. Note that the delay of the first spikes in each burst, the ones occurring after an inter-burst interval of 650 ms, increases disproportionately. This corresponds to the large delays at small $F_{inst}$ as seen in *Figure 2B* for the FTS effect during *Poisson*

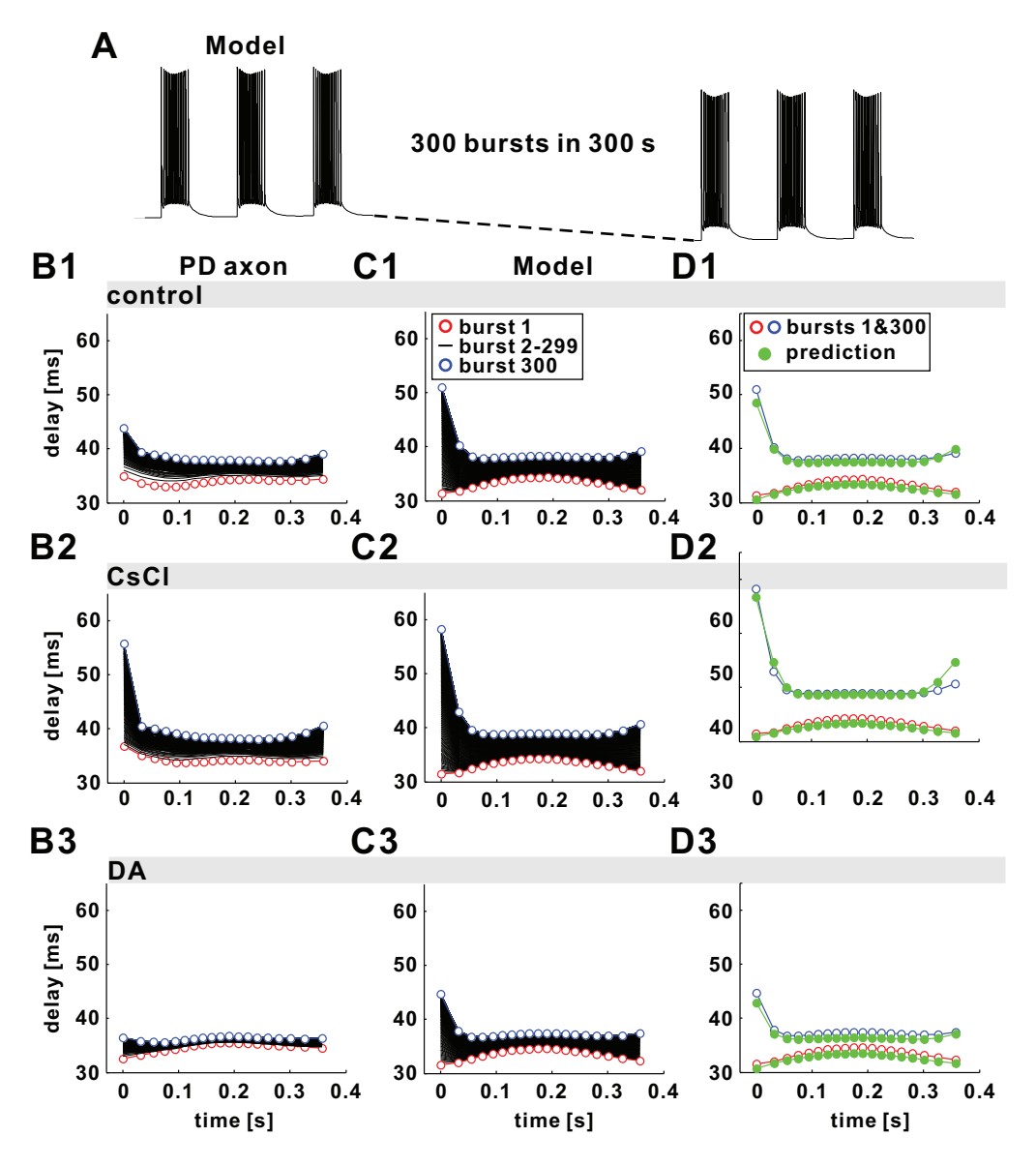

**Figure 7.** The fast timescale effect observed in the burst stimulations can be predicted by the paired-pulse stimulation results. (A) The model responses to 300 parabolic burst stimuli (350 ms duration; 1 s period) over 300 s show a drop in baseline membrane potential. (B1–B3) Experimental results of conduction delay of spikes in each burst shown, superimposed as a function of time from the burst onset. Data of the first (red) and the 300th (blue) burst are colored. (C1–C3) Model replication of the experimental results, with different $I_h$ levels (see *Tables 1* and *2* for model parameters). (D1–D3) Simulation results of the first and the 300th burst in panel *C1-C3* are predicted by estimating history-dependence using paired-pulse simulations (see *Figure 6*). Paired-pulse simulations were done with a constant low or high $I_{pump}$ level, equal, respectively, to the $I_{pump}$ mean value during the first and the 300th burst, fit with polynomial functions and used to obtain the plotted predictions (green).

stimulation. This increase in delay of the first spike in the burst was even more pronounced in the model (*Figure 7C1*).

In exploring spike conduction delay under natural bursting conditions, we also considered the effect of neuromodulation on history-dependence. As stated above, conduction delay in the PD axon is affected by the level of $I_h$, which is modulated by DA (*Ballo et al., 2010, 2012*). *Figure 7B2 and B3* show that both the slow increase in mean delay and the variability of

delay are more severe when $I_h$ was blocked by CsCl, and less severe when $I_h$ was increased by application of DA. In the model, these effects were mimicked when $I_h$ was blocked (*Figure 7C2*) or increased (*Figure 7C3*). As with *Poisson* stimulations, the increase in the conduction delay of each subsequent burst was because of the slow increase of $I_{pump}$ with stimulation (not shown), leading to a more hyperpolarized resting potential. This effect was counteracted to different degrees by different levels of $I_h$.

As with the FTS effect during *Poisson* stimulations, we examined whether the nonlinear relationships between conduction delay and time, seen in the burst stimulation of the biophysical model, can be predicted with paired-pulse stimulations at different levels of $I_{pump}$. A comparison of the paired-pulse estimates of these relationships for the first and last bursts (1 and 300) is shown in *Figure 7D1–3* for the three conditions: control, $I_h$ block and DA. The level of $I_{pump}$ was set to the mean value of the dynamical $I_{pump}$ at the time of burst 1 or the time of burst 300. The estimates for the paired-pulse stimulations shown in these panels were obtained from the polynomial fits (as in *Figure 6B*) and the ISIs of individual spikes. The nonlinear relationship between conduction delay and spike number was captured in all cases.

## Predicting the history-dependence of conduction delay

In addition to exploring the history-dependence of conduction delay at a phenomenological level, we also investigated which ionic current dynamics could account for this history-dependence. In the PD axon, delay is not a simple linear function of spike trajectory measures like peak voltage, trough voltage, or duration (*Ballo et al., 2012*). A possible reason for this is the dependence of conduction velocity on changes in total membrane conductance, which because of opposing inward and outward currents are not necessarily apparent in voltage traces (*Bucher and Goaillard, 2011*). A number of theoretical studies have provided equations to describe conduction velocity of a single isolated spike. The Matsumoto-Tasaki equation provides an estimate of velocity as a function of the total conductance level at the peak of the spike (see Materials and methods), and has been verified experimentally for single spikes in the squid giant axon (*Matsumoto and Tasaki, 1977*; *Tasaki and Matsumoto, 2002*; *Tasaki, 2004*). If the history-dependence of conduction delay is because of changes in the total conductance level, the Matsumoto and Tasaki equation should be able to predict the STS and FTS effects seen in our simulations. We therefore used this equation to estimate the delay of each spike in the 300 s, 10 Hz *Poisson* stimulation of our model (as shown in *Figure 2C, D*).

Although the Matsumoto and Tasaki equation provided a good first-order estimate of conduction delay, it did not capture the STS effect in the model axon (*Figure 8A1*). Additionally, for the FTS effect, this equation only showed a decrease of delay at low $F_{inst}$ and did not capture the variability (*Figure 8A2*). In general, the overall ability of the Matsumoto and Tasaki equation to predict the conduction delay in our model was poor (*Figure 8A3*).

An alternative method for estimating conduction velocity in the Hodgkin-Huxley model axon is provided by the Muratov equation (*Muratov, 2000*). This estimate uses the values of the $I_{Na}$ inactivation variable $h_{Na}$ and the $I_{Na}$ activation rate $\alpha_m = m_\infty / \tau_m^{Na}$, both evaluated at rest (see Materials and methods). *Figure 8B* shows a comparison of the delays in the *Poisson* stimulation of our model and the predicted delays using the Muratov equation. The predictions of delay by the Muratov equation were qualitatively similar to the simulation results for the STS effect, in that they showed an increase of $D_{mean}$ and CV-D. However, the increases in these two factors were limited and the equation failed to provide a quantitatively accurate prediction (*Figure 8B1*). Additionally, for the FTS effect, the increase of delay with larger values of $F_{inst}$ was not captured by this equation (*Figure 8B2*). Therefore, the overall ability of this equation to predict the conduction delays in the model axon was also poor (*Figure 8B3*). In conclusion, although these equations provide good predictions for conduction delay of an isolated spike, neither could accurately predict the short- and long-term history-dependence of conduction delay of the model axon.

## Sensitivity of the STS and FTS history-dependence of conduction delay to model parameters

Because previously published equations failed to predict accurately the history-dependence of conduction delay, we used an empirical approach to explore which model parameters contributed most

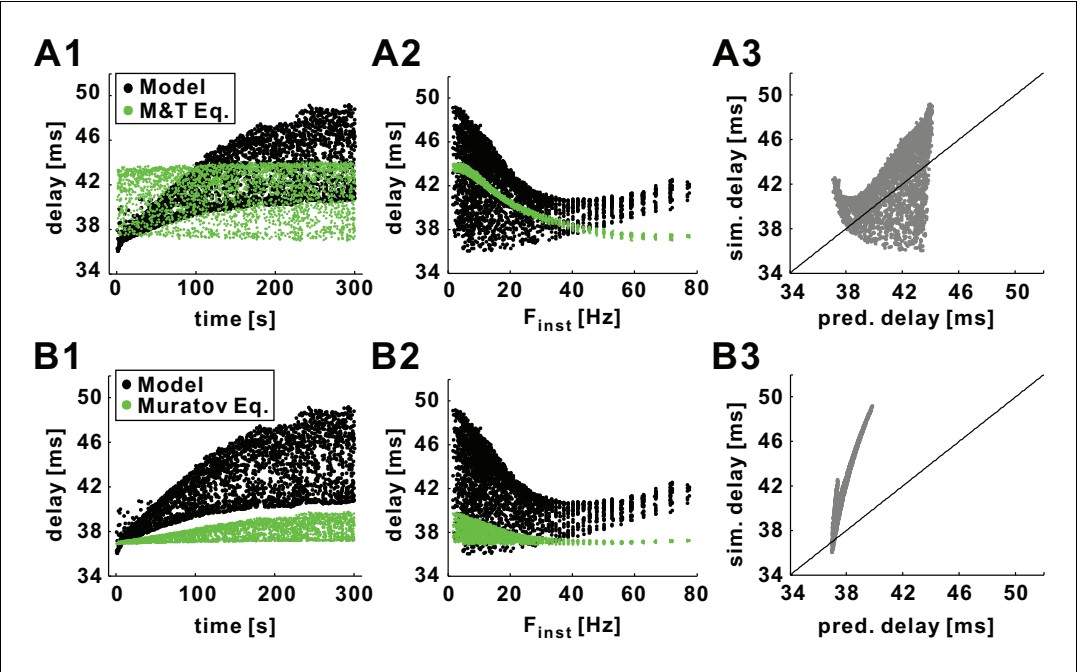

**Figure 8.** Predictions of history-dependence of conduction delay by previously published equations of action potential velocity. (**A1**) Conduction delays of the biophysical model in response to 10 Hz *Poisson* stimulation and delays predicted by the Matsumoto and Tasaki equation (***Tasaki and Matsumoto, 2002***). (**A2**) Data in panel A1 plotted versus $F_{inst}$. (**A3**) Simulation delay versus delay predicted by Matsumoto and Tasaki equation. (**B1–B3**) Comparison between simulation delays in the biophysical model (as in *A1-C1*) and the delays predicted by the Muratov equation (***Muratov, 2000***). Diagonal lines are *y=x*.

to this history-dependence. Our model included five distinct ionic currents, each of which depended on a maximal conductance and reversal potential and, with the exception of $I_{Leak}$, on parameters that determine the activation kinetics and, in the cases of $I_{Na}$ and $I_A$, inactivation kinetics. As a first step in understanding the contribution of these parameters to history-dependence of conduction delay, we examined the sensitivity of STS and FTS effects to changes in these parameters (see Materials and methods). The sensitivity was measured by examining how the different attributes that describe the STS and FTS effects (*Figure 9A*) depend on small changes in the parameter values. For the STS effect, these attributes were $D_{mean}$ and CV-D. For the FTS effect, we used quadratic fits to determine the minimum delay ($D_{min}$), the frequency ($F_{min}$) at which $D_{min}$ occurred, and the curvature of the fit at this point ($\kappa_{min}$).

Note that a small sensitivity value does not imply that the model attribute is not dependent on the parameter, but rather that small changes in the parameter do not affect that attribute strongly or monotonically. The large effect of $I_{pump}$ on conduction delay and history-dependence seen in *Figure 3* was because of several-fold changes in pump activation over several minutes. In contrast, the comparatively small (±5% and±10%) changes of $I_{pump}$ used here had little effect on sensitivity values (not shown). For all other parameters, we performed the sensitivity analysis at two different constant values of $I_{pump}$ corresponding to the mean $I_{pump}$ value at minute 1 (red) or 5 (blue) of the *Poisson* stimulation.

The primary result of the sensitivity analysis was that the parameters which produced the largest variations in any of the attributes of the STS (*Figure 9B and C*) or FTS (*Figure 9D–9F*) effect were those directly involved in spike generation, that is, the parameters of $I_{Na}$, $I_{Kd}$ and $I_{Leak}$. Although, as described above, $I_h$ contributes greatly to the history-dependence of conduction delay, the different attributes of history-dependence were not greatly sensitive to the parameters of $I_h$, nor to that of the transient potassium current $I_A$.

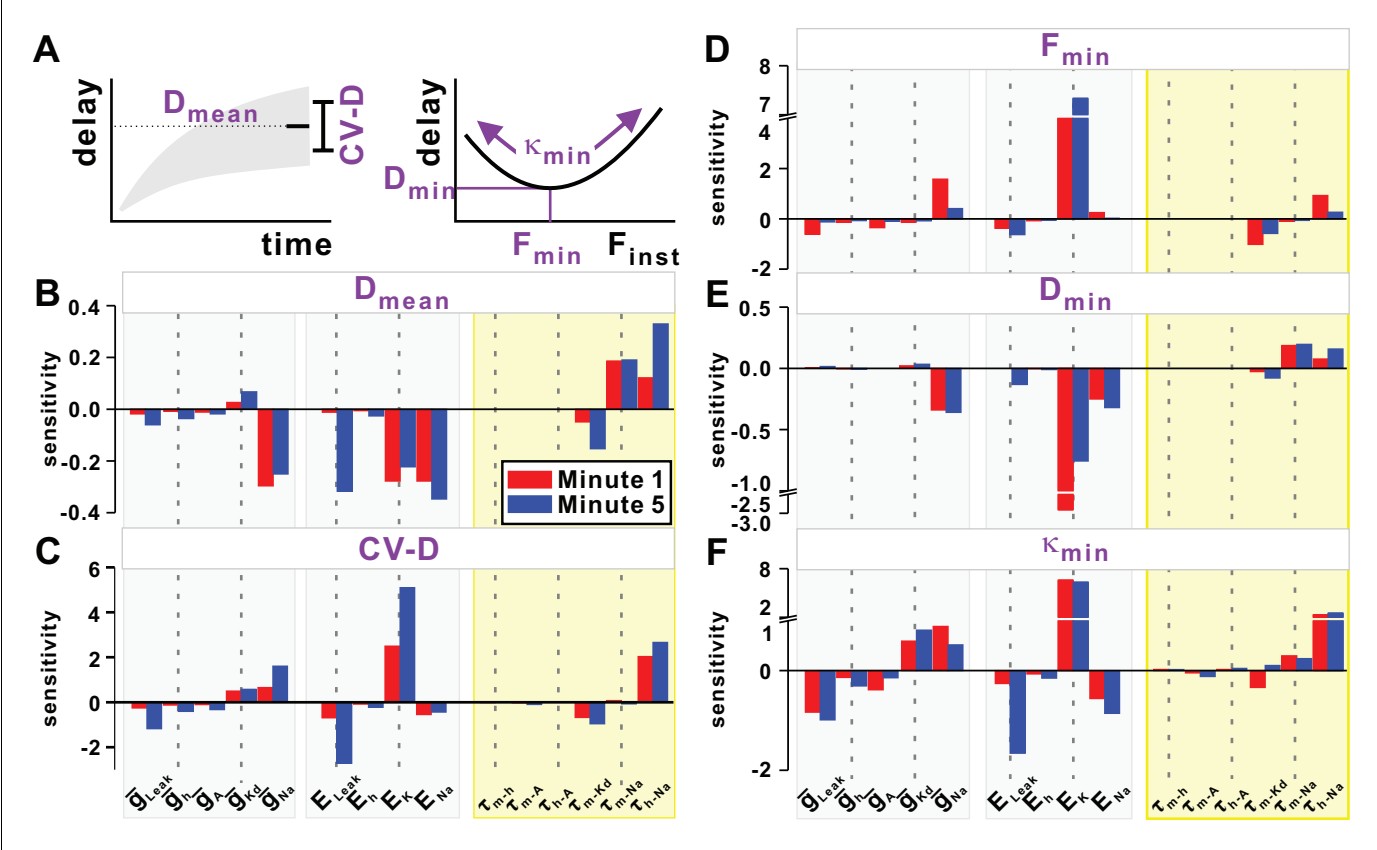

**Figure 9.** Sensitivity of the slow (STS) and fast timescale (FTS) effects to the model parameters. (A) Top left inset schematically shows the mean ($D_{mean}$) and variability of conduction delay (CV-D = coefficient of variation), measured in the last 20 s of simulation. Top right inset schematically shows the minimum delay point ($F_{min}$, $D_{min}$) and the curvature $\kappa_{min}$ of the quadratic fit at this point. (B–C) STS effect. Sensitivity of $D_{mean}$ and CV-D to model parameters in the first and five minute of a 10 Hz *Poisson* stimulation. (D–F) FTS effect. Sensitivity of $F_{min}$, $D_{min}$ and $\kappa_{min}$ to model parameters in the first and fifth minute of a 10 Hz *Poisson* stimulation.

The value of $D_{mean}$ (*Figure 9B*) strongly and negatively correlated with the equilibrium potential of K$^+$ ($E_K$) and the leak reversal potential ($E_{Leak}$). This sensitivity was most likely because of the contribution of $E_K$ and $E_{Leak}$ to the resting membrane potential. $D_{mean}$ also strongly and negatively depended on the equilibrium potential of sodium ($E_{Na}$) and its maximum conductance ($\bar{g}_{Na}$). This is consistent with the predictions of the Matsumoto-Tasaki equation, because an increase in either of these parameters results in an increase in $g_{Na}$, and therefore total conductance. Additionally, $D_{mean}$ strongly and positively depended on the activation and inactivation time constants of $I_{Na}$. The sensitivity of CV-D to model parameters was not as consistent as that observed for $D_{mean}$. Several parameters produced different ($E_K$, $E_{Leak}$, $\bar{g}_{Na}$, $\tau_h^{Na}$) effects on CV-D, when the $I_{pump}$ value was low or high (*Figure 9C*).

Of the FTS attributes, $F_{min}$ was sensitive primarily to $E_K$ (*Figure 9D*). Although it is difficult to gain a clear intuition on how different parameters affect $F_{min}$, this observation indicates that the fastest spikes can be obtained at a higher frequency if $E_K$ is shifted to a more depolarized value. Not surprisingly, the dependence of $D_{min}$ on the model parameters (*Figure 9E*) was similar to that of $D_{mean}$. The non-monotonic dependence of delay on the instantaneous stimulus frequency was captured primarily by the curvature of the quadratic fit. A larger curvature $\kappa_{min}$ implies a larger nonlinearity and therefore a larger difference between conduction delays at different $F_{inst}$ values. $\kappa_{min}$ is negatively dependent on $E_{Leak}$, $\bar{g}_{Leak}$ and $E_{Na}$, but positively on $\bar{g}_{Kd}$, $\bar{g}_{Na}$ $E_K$ and $\tau_h^{Na}$ (the time constant of $I_{Na}$ inactivation, *Figure 9F*). These dependencies are consistent, but the reasons underlying their effect on $\kappa_{min}$ are not obvious.

The sensitivity analysis included two types of parameters: those with constant values (maximal conductances and reversal potentials), and those whose values depend on the membrane potential and can change with time (ionic current time constants). The latter parameters can change in value for each spike and therefore are the only ones that can potentially contribute to the history-dependence of conduction delay. Within this category of parameters, the FTS and STS effects showed the highest sensitivity to the time constants of $I_{Na}$.

### History-dependence of conduction delay is dominated by two rate constants of the Na$^+$ channel kinetics

The dependence of STS and FTS effects on the gating time constants of $I_{Na}$ activation and inactivation, $\tau_m^{Na}(V)$ and $\tau_h^{Na}(V)$, led us to explore whether the factors that determine these time constants were responsible for the history-dependence of conduction delay.

$\tau_m^{Na}(V)$ and $\tau_h^{Na}(V)$ are determined by four rates (*Figure 10* inset): the opening and closing rates of activation, $\alpha_m(V)$ and $\beta_m(V)$, and the opening and closing rate of inactivation, $\alpha_h(V)$ and $\beta_h(V)$. We therefore examined how these rates changed for each spike during the *Poisson* stimulation. Because the opening rate is relevant at the onset of the spike and the closing rate during the spike itself, we evaluated $\alpha_m$ and $\alpha_h$ at the spike trough voltage ($V_T$, the voltage minimum right before the spike),

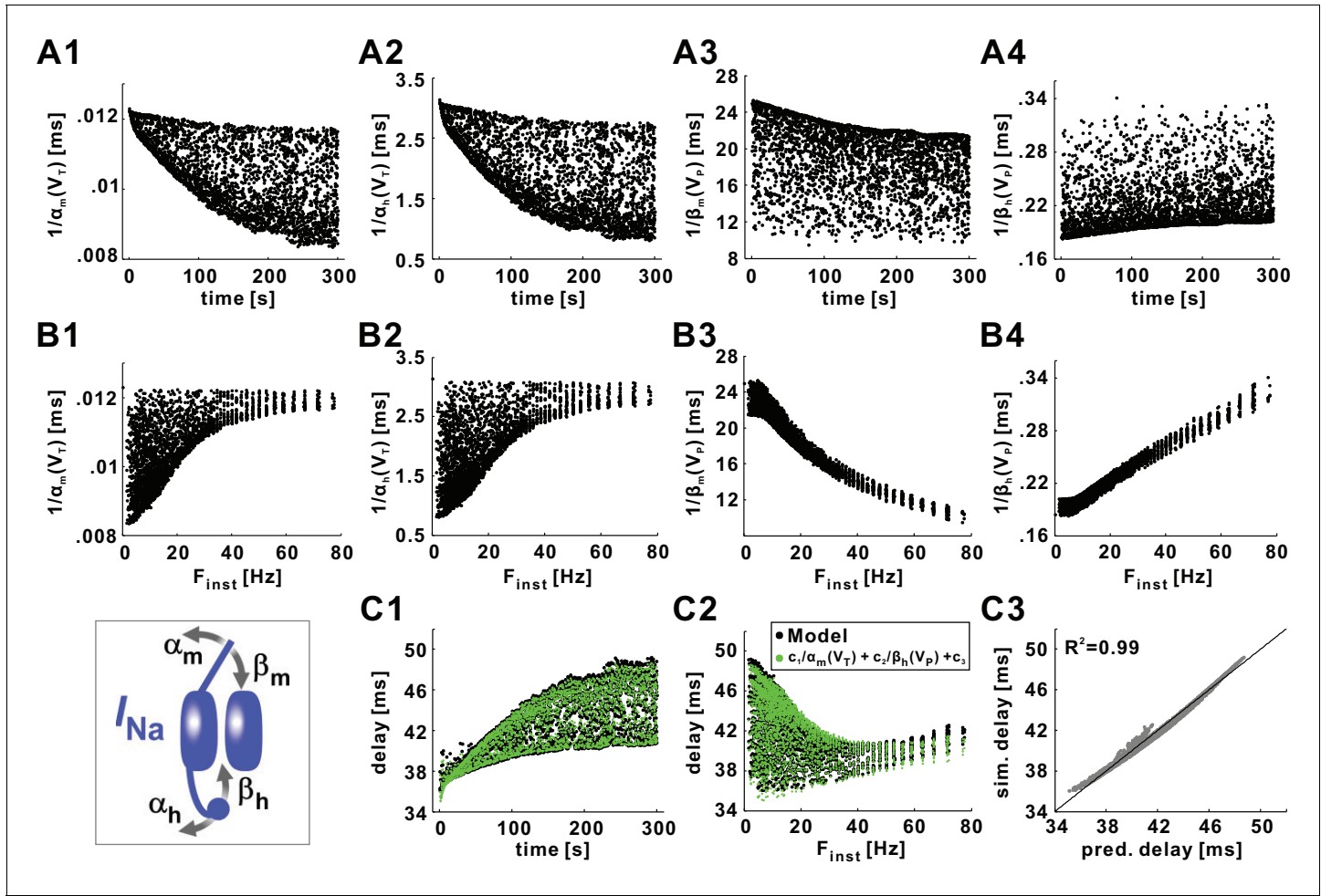

**Figure 10.** Conduction delay is determined by the $I_{Na}$ activation and inactivation variables, evaluated at the action potential trough ($V_T$) or peak ($V_P$) voltage. (A1–A2) Reciprocal of the $I_{Na}$ activation rate, evaluated at $V_T$. (A3–A4) Reciprocal of the $I_{Na}$ inactivation rate, evaluated at $V_P$. (B1–B4) Data in panel A plotted as a function of $F_{inst}$. (C1) Conduction delay of the model compared with the predicted delay of $c_1/\alpha_m(V_T)+c_2/\beta_h(V_P)+c_3$ ($c_1 = -3361.48$, $c_2 = 32.80$, $c_3 = 70.14$). (C2) Data in panel C1 plotted as a function of $F_{inst}$. (C3) Simulation delays compared with predicted delays from the empirical equation ($R^2=0.99$). The line is $y=x$. The inset is the schematic of the fast sodium channel.

and $\beta_m$ and $\beta_h$ at the spike peak voltage ($V_P$). In order to compare the changes in these parameters directly with those of conduction delay, we plotted the reciprocal values of the parameters, which have units of time.

All four parameters changed substantially during the *Poisson* stimulation and showed both STS and FTS effects (*Figure 10A and B*), but no single parameter followed exactly the STS and FTS effects on conduction delay. However, both $1/\alpha_m(V_T)$ and $1/\alpha_h(V_T)$, the (reciprocals of the) rates of activation and inactivation gates opening at the spike onset, showed an STS effect that was qualitatively similar to that of conduction delay (*Figure 10A1 and A2*); yet the STS effect for these parameters was decreasing rather than increasing, and neither parameter captured the non-monotonic FTS effect (*Figure 10B1 and B2*). In contrast, neither $1/\beta_m(V_P)$ nor $1/\beta_h(V_P)$, the (reciprocal) rates of activation and inactivation gates closing at the spike peak, qualitatively captured the increase in variance over time of the STS effect very well (*Figure 10A3 and A4*), and both these parameters showed a monotonic FTS effect (*Figure 10B3 and B4*). In summary, no single parameter could predict the overall time course of changes in conduction delay.

We next examined whether any combination of two parameters would better capture the STS and FTS effects. There are six possible combinations of two parameters out of these four. We fit the model STS and FTS effects with all six combinations. That is, for any combination of two parameters $x$ and $y$ out of the four, we fit the *Poisson* stimulation data of *Figure 2C–D* with the equation

$$d_{est} = c_1 x + c_2 y + c_3.$$

The coefficients $c_i$ ($i$ = 1,2,3) were determined with a Powell's optimization method and the estimated delay was compared with the simulation results. We found that five of the combinations produced a fit to the model delay values that had a coefficient of determination $R^2$ of >0.9, but only two of these qualitatively captured both the STS and FTS effects seen in *Figure 2C–F* (data not shown). These combinations were $1/\alpha_h(V_T)$, $1/\beta_h(V_P)$, and $1/\alpha_m(V_T)$, $1/\beta_h(V_P)$. Of these, the better fit was obtained from the combination that included both the activation and inactivation gating variables ($R^2$ = 0.99 compared to 0.97). We therefore used the following empirical equation as the best estimate for the model conduction delay:

$$d_{est} = \frac{c_1}{\alpha_m(V_T)} + \frac{c_2}{\beta_h(V_P)} + c_3.$$

*Figure 10C1 and C2* show that this equation captures both STS and FTS effects very well. *Figure 10C3* shows the high accuracy with which this equation predicts the conduction delays in the model ($R^2$ = 0.99). Although, for brevity, we do not show the dependence of conduction delay on $\alpha_m(V)$ and $\beta_h(V)$, scaling these functions had a direct effect on the conduction delay. We therefore conclude that the opening rate of the activation gate of the sodium channel right before the spike, and the closing rate of the inactivation gate of the sodium channel at the peak of the spike are good proxies for, and likely important determinants of, the spike conduction velocity in axons.

## History-dependence of conduction delay can be predicted from the action potential voltage trajectory

The equation to predict conduction delay from activation and inactivation variables served to gain insight into underlying mechanisms. However, it is of limited value for experimental work, because activation and inactivation variables cannot be measured during ongoing spiking. We therefore asked if there are good enough correlates of activation and inactivation variables to be found in the voltage trajectory of spikes that would allow the prediction of delay. Single spike measures like peak or trough voltages are poor predictors of delay in the PD axon (*Ballo et al., 2012*), but the dependence of delay on channel gating at both trough and peak shown here suggests that more than one measure has to be considered. We noticed that although $\alpha_m$ and $\beta_h$ and are nonlinear functions, in the range of membrane potentials restricted to the trough or peak of the spike ($V_T$ and $V_P$), both $\alpha_m(V_T)$ and $\beta_h(V_P)$ were almost linear functions of their respective variables. We therefore treated $V_T$ and $V_P$ as linear approximations of $1/\alpha_m(V_T)$ and $1/\beta_h(V_P)$, respectively, and used them instead in our equation:

$$d_{est} = \frac{c_1}{V_T} + \frac{c_2}{V_P} + c_3$$

With the same optimization process as before, the coefficients were determined for the *Poisson* stimulation data of the model axon. The resulting prediction of conduction delay captured the history-dependence of delays with 99% accuracy (*Figure 11A1–A3*). Furthermore, the equation with the same coefficient values predicted the delays in a novel set of *Poisson* stimulation data with the same accuracy (not shown).

We also determined the contribution of each variable to the fit. To this end, we examined how well the simulated delays can be fit using only $1/V_T$ or only $1/V_P$ as the variable. In these fits, the other variable ($V_P$ or $V_T$) was set to its mean value measured during the 300 s *Poisson* stimulation. The results of these fits are shown in *Figure 11B and C*. Neither variable perfectly captured the STS effect although this effect was better captured with fits using only $1/V_T$ (*Figure 11B1*). This result is consistent with the sensitivity of $D_{mean}$ to $\tau_m^{Na}$ (see *Figure 9B*), because a larger $\tau_m^{Na}$ results in a slower

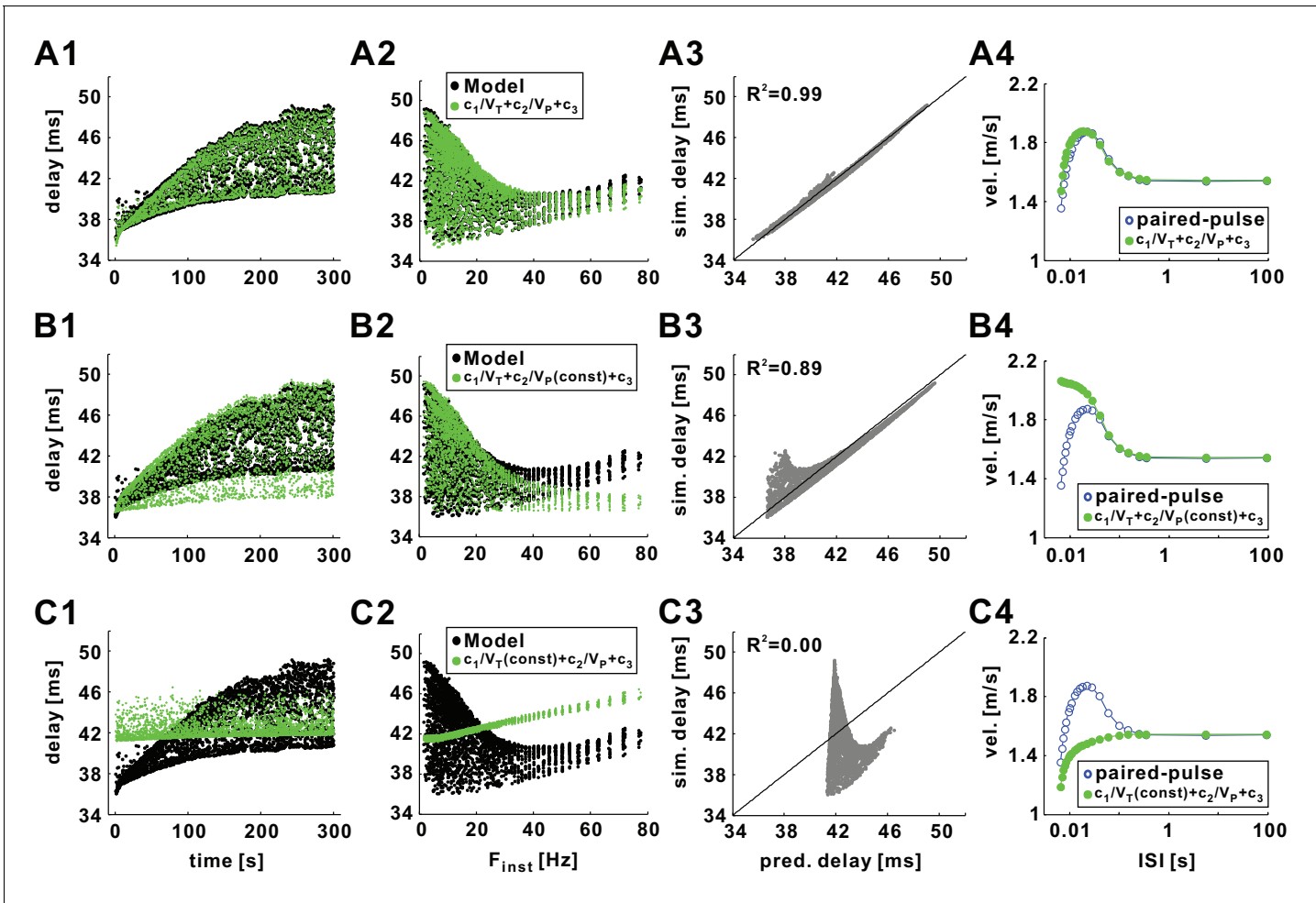

**Figure 11.** Conduction delay can be almost perfectly predicted by the trough and peak voltages of the action potentials. **(A1)** Superimposed graph of conduction delays of the biophysical model (black) in response to 10 Hz *Poisson* stimulation and the predicted delays from the nonlinear regression equation: $d = c_1/V_T + c_2/V_P + c_3$ ($c_1 = 4072.88$, $c_2 = 302.36$, $c_3 = 82.94$; the same values of $c_1$-$c_3$ are used in all panels). **(A2)** Data in panel *A1* plotted as a function of $F_{inst}$. **(A3)** Simulation delays (training data) compared with predicted delays ($R^2 = 0.99$). **(A4)** Conduction delays of the biological model (blue) in response to paired-pulse stimulation (same as in *Figure 6A*) and the predicted delays from the regression equation in panel *A1*. **(B1–B3)** Simulation results (same as in *A1-A3*) compared with predicted delays from the single variable regression: $d = c_1/V_T + c_2/V_P$ (mean) $+c_3$, where $V_P$ (mean) is the mean value of $V_P$. **(B4)** Simulation results (same as in *A4*) compared with predicted delays from the single variable regression: $d = c_1/V_T + c_2/V_P$ (cond) $+c_3$, where $V_P$ (cond) is the value of $V_P$ of the conditioning pulse. **(C1–C3)** Simulation results (same as in *A1-A3*) compared with predicted delays from the single variable regression: $d = c_1/V_T(mean)+c_2/V_P +c_3$, where $V_T$ (mean) is the mean value of $V_T$. The lines in *A3, B3, and C3* are y=x. **(C4)** Simulation results (same as in *A4*) compared with predicted delays from the single variable regression: $d = c_1/V_T(cond)+c_2/V_P +c_3$, where $V_T$ (cond) is the value of $V_T$ of the conditioning pulse.

opening rate of $I_{Na}$ activation variable at $V_T$. Similarly, for the FTS effect, neither variable alone captured the non-monotonic relationship of delay with $F_{inst}$; however, the decrease of delay with $F_{inst}$ was best approximated by $1/V_T$ (*Figure 11B2*) whereas its increase was best approximated by $1/V_P$ (*Figure 11C2*). Therefore, although neither $1/V_T$ nor $1/V_P$ can accurately predict conduction delay on their own, the multivariate linear regression using both $1/V_T$ and $1/V_P$ provides an accurate prediction of history-dependence of conduction delay in the biophysical model.

Our description of delay changes suggests that they are dominated by the $I_{Na}$ activation state before the spike and the inactivation state at the peak, and well reflected in the voltage values at these time points (*Figure 10*). We therefore predicted that, while other conductances can influence the gating state of $I_{Na}$ through their effect on membrane potential, the recovery cycle should be accurately predicted by our equation describing the dependence of delay on trough and peak voltages. To test this prediction, we used the estimated delays from our equation to predict the results of the paired-pulse stimulations shown in *Figure 6A* (with $I_{pump,HI}$). As described above, the coefficients $c_i$ ($i$ = 1,2,3) were independent of the stimulation pattern once the model axon was fixed. We therefore used coefficients obtained by a 1 min *Poisson* stimulation protocol to predict the conduction velocities of the spikes in the paired-pulse stimulation. With these coefficients and the $V_T$ and $V_P$ values, the conduction velocities were accurately predicted for both levels of $I_{pump}$ (*Figure 11A4*).

The extent to which either $V_T$ or $V_P$ can separately predict delays during *Poisson* stimulation was discussed above (for *Figure 11B and C*). We also tested the extent to which $V_T$ and $V_P$ contributed to the different periods of the recovery cycle. We restricted this analysis to the paired-pulse simulation results for high levels of $I_{pump}$ (*Figure 11B4 and C4*). In order to understand the contribution of $V_T$ to changed excitability at different intervals, we set $V_P$ to a constant value equal to the trough voltage of the conditioning spike in the paired-pulse stimulation. Using the coefficients from the *Poisson* stimulation and $V_T$ from the paired-pulse stimulation, the predicted conduction velocity decreased with increasing ISIs in the range of relative refractory and supernormal periods (*Figure 11B4*). Therefore, changes in $V_T$ captured the supernormal period. The decrease of velocity was consistent with the results for *Poisson* stimulations shown in *Figure 11B2*: predicted conduction delay decreased with $F_{inst}$ if only $V_T$ was used.

To capture the contribution of $V_P$, we fixed the value of $V_T$ to a constant equal to the peak voltage of the conditioning spike in the paired-pulse stimulation. Again, using the same coefficients and the $V_P$ values from the paired-pulse stimulation, we found that the predicted conduction velocity monotonically increased with ISI (*Figure 11C4*). Thus the changes in $V_P$ captured the relative refractory period, which was consistent with the results shown in *Figure 11C2*: predicted conduction delay increased with $F_{inst}$ if only $V_P$ was used.

## History-dependence of experimentally measured conduction delay can be predicted from the voltage trajectory of spikes

Because both $V_T$ and $V_P$ are readily measured from experimental intracellular recordings (*Ballo et al., 2012*), we asked whether our equation can be used to predict the history-dependence of conduction delays in the PD axon. We used conduction delays measured with a *Poisson* stimulation at 10 Hz and fit the delay values with our equation. We then used this equation to predict the delay values of a novel dataset produced with the same stimulation rate. The results showed that we could predict both the STS and the FTS effects of the experimental data relatively accurately (*Figure 12A–B*). The fit produced a coefficient of determination of 0.87, indicating that 87% of the variance of the experimental delays can be predicted by this equation (*Figure 12C*). The history-dependence of conduction delay in experiments can therefore be predicted from the spike voltage trajectories without any need for computational modeling.

## Discussion

The classic description of ionic mechanisms underlying spike generation and conduction was based on only two voltage-gated currents to describe the membrane behavior of the squid giant axon (*Hodgkin and Huxley, 1952a*, *1952b*). This relatively simple model has dominated the common perception of axonal spike propagation, even though ionic mechanisms in most axons are more complex, as multiple currents with very different properties can be present (*Krishnan et al., 2009*; *Bucher and Goaillard, 2011*). Later theoretical work examined in detail the properties determining

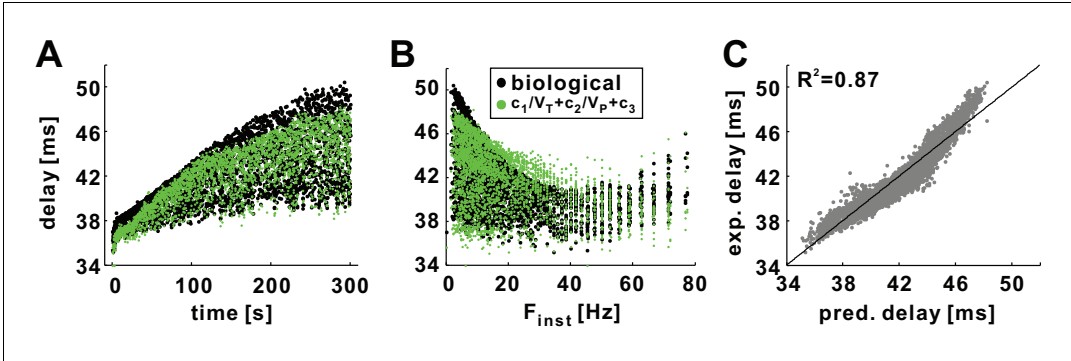

**Figure 12.** The history dependence of conduction delay in the biological PD axon can be predicted by the empirical equation without need for computational modeling. (**A**) Conduction delays in the biological PD axon in response to a 5 min, 10 Hz, *Poisson* stimulation (black; same as in *Figure 2A*) compared with delays predicted by empirical equation ($c_1$=4817.75, $c_2$ = 500.5, $c_3$ = 105) using $V_T$ and $V_P$ values from intracellular recordings (not shown). (**B**) Data in panel *A* plotted as a function of $F_{inst}$. (**C**) Experimental delays plotted versus the predicted delays ($R^2$ = 0.87). The line is $y=x$.

conduction velocity (*Muratov, 2000*; *Tasaki, 2004*), but only for single spikes and considering only a limited set of properties. We show that these models fail to capture the history-dependence of conduction velocity during repetitive activity. Such history-dependence is evident in findings that changes in axonal excitability and conduction velocity can occur at timescales far exceeding single inter-spike intervals, and can substantially alter the temporal pattern of spikes during propagation (*Weidner et al., 2002*; *Ballo et al., 2012*). It is important to understand the mechanisms underlying history-dependence of spike propagation, because changes in temporal patterns potentially play an important role in shaping the neural code (*Izhikevich, 2006*; *Bucher and Goaillard, 2011*; *Bucher, 2015*).

## The dependence of conduction delay on slow activity-dependent hyperpolarization

We describe the history-dependence of conduction delay in a biophysically realistic model axon designed to match properties of the PD motor axon of the crustacean stomatogastric nervous system. The model captures well how delay changes as a function of prior spiking in this axon, at two different timescales (*Ballo et al., 2012*)(*Figure 2*).

At the slow timescale (STS effect), repetitive activity leads to cumulative hyperpolarization and conduction slowing. Over the course of minutes, mean delay and variability of delay increases. We show that the STS effect is due to the Na$^+$/K$^+$ pump (*Figure 3*). It has been proposed for many axons that slow hyperpolarization and slowing of conduction is caused by this pump, which is activated by the massive Na$^+$ influx during repetitive spiking and causes a net deficit of positive charge (*Van Essen, 1973*; *Gordon et al., 1990*; *Bostock and Bergmans, 1994*; *Robert and Jirounek, 1994*; *Vagg et al., 1998*; *Baker, 2000*; *Kiernan et al., 2004*; *Moldovan and Krarup, 2006*; *Scuri et al., 2007*). Direct experimental evidence of the involvement of the pump in activity-dependent dynamics for the PD axon is missing because of the difficulty that pharmacological block of the pump also interferes with its overall role in maintaining a functional membrane potential (*Ballo and Bucher, 2009*). However, other slow outward currents have not been identified in the PD axon (Bucher, unpublished results).

We nevertheless considered a slowly activating outward current as an alternative mechanism because many axons express slow K$^+$ channels, e.g. K$_v$7 channels that produce M-type currents (*Dubois, 1981*; *Baker et al., 1987*; *Eng et al., 1988*; *Devaux et al., 2004*; *Schwarz et al., 2006*; *Vervaeke et al., 2006*; *Buniel et al., 2008*). Such currents are thought to contribute to hyperpolarization and reduced excitability and may have cumulative effects during repetitive activity (*Baker et al., 1987*; *Taylor et al., 1992*; *Stys and Waxman, 1994*; *Miller et al., 1995*; *Lin et al., 2000*; *Burke et al., 2001*; *Schwarz et al., 2006*). An interesting difference between the pump

current and currents due to ion channel conductances is that the pump current does not depend on a driving force. Equivalently, an ionic current would change the input conductance of the neuron, whereas the pump current does not. We found that even with a low activation threshold and slow kinetics, the slow voltage-gated K$^+$ current cannot cause substantial activity-dependent hyperpolarization because an $E_K$ relatively close to the resting membrane potential rendered the driving force in between spikes too small for substantial summation of current (*Figure 4*). The same would be true for Cl$^-$ conductances and $E_{Cl^-}$. Yet, this slow K$^+$ current reduced the peak voltage of action potentials and reproduced the effect of the pump current in increasing the mean delay D$_{mean}$, but not the variability CV-D.

It should be noted, however, that we used an $E_K$ value ($-70$ mV) in our axon model that is about 10 mV less negative than usually assumed for soma recordings of STG neurons (*Golowasch and Marder, 1992*). A sufficiently slow voltage-gated K$^+$ with $E_K$ substantially lower than the resting potential could possibly reproduce the effect of the pump current in producing the STS effect. We chose the $E_K$ value near the resting potential because spikes in PD axon recordings show an absence of fast after-hyperpolarization (undershoot), and we could only replicate this in the model with a less negative $E_K$. An alternative approach would have been to model $E_K$ as a dynamic variable and allow it to transiently change to more depolarized values over the course of a single spike. This type of dynamic $E_K$ was used in a model of the squid giant axon, which assumed a significant temporary extracellular K$^+$ accumulation, and was successful in rendering a more realistic voltage trajectory of repolarization and undershoot (*Clay, 1998*). As the PD axon does not actually express slow K$^+$ currents, we did not pursue this further. However, our results demonstrate that the increase of variability of delay was dependent on hyperpolarization, as a mere increase in conductance only increased the mean delay.

The dependence of the STS effect on hyperpolarization is further highlighted by the role of $I_h$ (*Figure 5*), which is thought to play an important role in preventing spike failures in many axons because its inwardly rectifying properties can balance activity-dependent hyperpolarization (*Baker et al., 1987*; *Grafe et al., 1997*; *Soleng et al., 2003*; *Kiernan et al., 2004*; *Baginskas et al., 2009*; *Tomlinson et al., 2010*). In the PD axon, $I_h$ is increased several-fold by dopamine (*Ballo et al., 2010*), which substantially reduces hyperpolarization and history-dependence of conduction delay (*Ballo et al., 2012*). Our model provides a mechanistic explanation of our previous experimental observation that the balance of $I_h$ and pump current plays a dynamic role in setting how sensitive conduction is to the history of activity. This represents a level of importance in controlling temporal fidelity that goes beyond just preventing spike failures.

Previous studies have also used a modeling approach to explore the role of slow accumulation of ionic currents and concentrations in axons. The topics explored included explorations of Na$^+$ accumulation and the effects of the Na$^+$/K$^+$ pump in the coupled left-shift of the voltage-dependence of Na$^+$ channels due to axonal injury (*Boucher et al., 2012*; *Yu et al., 2012*; *Lachance et al., 2014*), the influence of Na$^+$ channel clustering on spike conduction (*Freeman et al., 2016*) and activity-dependent slowing of spike conduction (*Tigerholm et al., 2014*).

## The dependence of conduction delay on spike intervals

The model also captures well the PD axon's nonlinear and non-monotonic dependence of delay on instantaneous frequency (FTS effect), the manifestation of which is influenced by the STS effect. It has long been known that, with increasing spike intervals even well beyond the refractory period, excitability and conduction velocity go through an oscillation of decreases and increases, (*Bullock, 1951*; *Raymond, 1979*), collectively termed the recovery cycle. Such activity-dependent changes are often gauged with paired-pulse stimulations (*Bucher and Goaillard, 2011*), particularly for diagnostic purposes in the context of peripheral neuropathies (*Bostock and Rothwell, 1997*; *Bostock et al., 1998*; *Burke et al., 2001*; *Krishnan et al., 2009*). The dependence on spike interval changes dramatically when axons are stimulated with trains of conditioning pulses instead of simple paired pulses. In our model, prior activity can be mimicked by setting the pump current to different values (*Figure 6*). This suggests that FTS and STS effects are completely separated in their time dependence, meaning that delay depends on the last interval and the mean rate of prior activity, but not on the exact temporal pattern of the last few preceding spikes. This independence of exact prior pattern is further demonstrated by our finding that paired-pulse stimulation results predict delays from both *Poisson* (*Figure 6*) and burst (*Figure 7*) stimulations. A previous modeling study on

primary afferent C-fibers has also demonstrated an activity-dependent alteration of recovery cycles (*Tigerholm et al., 2014*).

## The dependence of conduction delay on Na$^+$ channel gating variables and other currents

Our results show that the dynamics of conduction delay can be accurately predicted from two parameters describing the state of the Na$^+$ conductance at two time points, the opening rate of the activation gate at the onset of the spike, and the closing rate of the inactivation gate at the peak (*Figure 10*). In comparison, two equations that accurately describe conduction delay of single spikes fail to capture the dynamics. The Matsumoto-Tasaki equation predicts conduction delay dependent on the change in total membrane conductance elicited by a spike (*Matsumoto and Tasaki, 1977*; *Tasaki and Matsumoto, 2002*; *Tasaki, 2004*). The Muratov equation includes Na$^+$ channel gating variables, but only at the rest state (*Muratov, 2000*). Both equations aim to predict absolute values of delay and include constant biophysical parameters such as diameter and membrane capacitance. In contrast, our empirical equation describes only how conduction delay changes as a function of activity. The magnitude of the delay changes and the absolute range of values are found by scaling the contribution of the two Na$^+$ gating rates with two constants and adding a third. These three constants have to be found empirically and may represent stand-ins for actual biophysical parameters.

Interestingly, the gating parameters at the trough potential before a spike and the peak voltage of the spike correlate linearly with these voltages, so that the voltages can be treated as stand-ins for the gating parameters in the empirical fit, which allows us to predict delay solely from voltage trajectories (*Figure 11*). This illustrates the important point that, while both our sensitivity analysis (*Figure 9*) and the empirical prediction of delays seem to suggest that Na$^+$ channel gating is the key factor, the rates used to predict delay are voltage-dependent and are therefore sensitive to the effect other ionic mechanisms have on the membrane potential trajectory.

It should be noted that the small level of sensitivity to the various currents is because sensitivity analysis only examines small changes in the parameters of these currents which would not greatly alter the membrane potential. However, some of these currents, as shown for the pump current and $I_h$, undergo large changes that have substantial effects on delay. Again, these mechanisms act at two different timescales. At the slow timescale, the balance of pump current and $I_h$ determines the overall level of hyperpolarization and thus affects Na$^+$ channel gating through the steady-state voltage-dependencies of the opening and closing rates. It should be noted that changes in baseline membrane potential overall can have ambiguous effects on conduction velocity. For example, slow hyperpolarization is often associated with slowing of conduction but can also increase removal of Na$^+$ channel inactivation and therefore increase velocity (*Vervaeke et al., 2006*; *De Col et al., 2008*).

At the fast timescale, K$^+$ currents play an important role in shaping the repolarization speed, after-potentials, and re-excitability, the extent of which critically depends on the gating time constants and temporal overlap with the Na$^+$ current (*Erisir et al., 1999*; *Baranauskas, 2007*; *Sengupta et al., 2010*). K$^+$ currents with different gating time constants may differentially affect both Na$^+$ channel activation and its inactivation. Of note is the common presence of inactivating and low-threshold K$^+$ currents in axons (*Krishnan et al., 2009*; *Bucher and Goaillard, 2011*; *Debanne et al., 2011*; *Bucher, 2015*), because their own activation and inactivation states will be critically dependent on slow timescale changes in baseline membrane potential and therefore change their effect on voltage trajectories and Na$^+$ channel gating. In the PD axon, relatively slow spike repolarization leads to summation even at moderate frequencies, and spike duration depends on $I_A$ activation and inactivation, which vary substantially with the baseline membrane potential (*Ballo and Bucher, 2009*).

## The relevance of the findings to other axons

The model presented here aimed to mimic, as faithfully as possible, the behavior of a well described unmyelinated invertebrate axon. However, it should be noted that the ionic mechanisms considered, like pump-mediated hyperpolarization, inward rectification through $I_h$, and the presence of both transient and sustained K$^+$ currents, are found in many axons across different phyla (*Bucher and Goaillard, 2011*). In contrast, diversity of axon properties within a single species or even a single

part of the nervous system can be very large, which makes the notion of a 'typical' axon appear not useful (*Bucher, 2015*). Nevertheless, spikes in the overwhelming majority of axons are driven by fast transient Na$^+$ currents responsible for their regenerative nature. It is plausible as a general mechanism that the gating state of these channels at the onset and peak of the spike acts as a critical determinant of conduction velocity. Different complements of additional voltage-gated or electrogenic transporter currents across axon types may lead to very different dependence of conduction velocity on preceding activity. Yet, we expect that their influence on the Na$^+$ channel gating variables, or trough and peak voltages as their proxies, would still serve to determine conduction velocity in each case. Consequently, once the peak and trough voltages of the action potential have been measured, the empirical equation introduced in this study would serve to accurately predict the conduction velocity, independent of the combinations of ionic mechanisms involved.

## Materials and methods

### The biophysical axon model

We constructed a conductance-based biophysical axon model to examine the role of different ionic currents in shaping the history-dependence of conduction delay. All simulations were done in NEURON (*Carnevale and Hines, 2006*). The model was based on standard cable equations for an unmyelinated axon, standard Hodgkin-Huxley type ionic conductances with different voltage- and time-dependencies, and a pump current representing the Na$^+$/K$^+$-ATPase:

$$\frac{a}{2R_i}\frac{\partial^2 V}{\partial x^2} = C_m \frac{\partial V}{\partial t} + \sum I_{ion} + I_{pump} + I_{app}^0(t).$$

The parameter $a$ is the radius (=5 μm) of the model axon, $R_i$ (=80 Ωcm) is the specific intracellular resistivity, $V$ is voltage, $x$ is the position along the axon, and $C_m$ (=1 μF/cm$^2$) is the membrane capacitance per unit area. The ionic currents ($\Sigma I_{ion}$), the Na$^+$/K$^+$ pump current ($I_{pump}$), and the stimulus current ($I_{app}^0(t)$) are described below. A specific membrane resistivity $R_m$ value of 8 KΩcm$^2$ was used to calculate the leak conductance, resulting in a passive length constant $\lambda$ = 1581 μm. Both $a$ and $\lambda$ were similar to the values found in the PD axon (*Ballo and Bucher, 2009*). The length of the model axon was set to 1 cm, and divided into 101 identical compartments for simulation. In order to apply the finite difference method to solve the model equations numerically, all compartments were assumed to be isopotential during the simulation process. All axon parameters are listed in *Table 1*.

### Ionic currents

Ionic currents used in the model are shown in *Figure 1A*. Different versions of the model contained different combinations of ionic currents (currents not included in all versions are shown in brackets):

$$\sum I_{ion} = I_{Na} + I_{Kd} + I_{Leak} + I_A \left[+I_h\right]\left[+I_{Ks}\right]\left[+I_{pump}\right].$$

In the simplest version, it included only the standard Hodgkin-Huxley-like leak ($I_{Leak}$), fast sodium ($I_{Na}$), and delayed-rectifier potassium ($I_{Kd}$) currents.

The PD axon expresses two additional voltage-gated ionic currents, a transient potassium current $I_A$, and a hyperpolarization-activated inward current $I_h$ (*Ballo and Bucher, 2009*; *Ballo et al., 2010*). $I_h$ is increased by dopamine (DA, *Figure 1A*) through a D$_1$-type receptor mechanism that increases cAMP levels (*Ballo et al., 2010*). We mimicked DA effects on $I_h$ by using different values of voltage half-activation and slope (*Table 2*) that were described experimentally (*Ballo et al., 2010*). However, these changes alone were not sufficient to produce the experimentally-observed effect of $I_h$ increase by DA on axonal delay, presumably because of differences in resting membrane potential between our model and the biological neuron. To match the biological results, we also assumed that DA increased $\bar{g}_h$ (*Table 1*). To mimic block of $I_h$, $\bar{g}_h$ was set to 0.

We also tested the effect of a slow potassium current ($I_{Ks}$). To this end, we introduced a conductance with the same voltage dependence as $I_{Kd}$, but a 5000 times slower time constant of activation.

All ionic currents were Hodgkin-Huxley type (*Hodgkin and Huxley, 1952b*) and were represented by equations of the following form:

$$I = \bar{g}m^p h^q (V - E_{rev})$$

$$\frac{dx}{dt} = \frac{x_\infty - x}{\tau_x}; x = m, h$$

where $\bar{g}$ is the maximum conductance, $E_{rev}$ is the reversal potential, $m$ and $n$ are the activation and inactivation variables, respectively, and $p$ and $q$ are non-negative integers. As stated above, gating parameters and $E_{rev}$ for $I_h$ were based on actual measurements in the PD axon (**Ballo et al., 2010**). $I_{Na}$ and potassium current parameters were based on common values and tuned to produce spike shapes that matched those in intracellular PD axon recordings (**Ballo and Bucher, 2009**). Of note in these recordings is the absence of an undershoot following a spike. We only managed to replicate this by setting the K$^+$ equilibrium potential $E_K$ to a fairly depolarized value of $-70$ mV (see Discussion). All gating parameters are listed in **Table 2**.

## The Na$^+$/K$^+$ pump

The Na$^+$/K$^+$ pump current was modeled in a simplified form as described previously (**Angstadt and Friesen, 1991**), and was governed by the following equation:

$$I_{pump} = \frac{I_{max,pump}}{1 + \exp\left(\frac{[Na^+]_{1/2} - [Na^+]_{in}}{[Na^+]_S}\right)}$$

where $I_{max,pump}$ is the maximum current, $[Na^+]_{in}$ the intracellular sodium concentration, $[Na^+]_{1/2}$ the concentration at which the pump is half active, and $1/[Na^+]_S$ describes the sensitivity of the pump to alterations of intracellular sodium concentrations. The rate of change for $[Na^+]_{in}$ was governed by:

$$\frac{d[Na^+]_{in}}{dt} = -\frac{I_{Na} + 3I_{pump}}{\alpha \cdot F \cdot Vol}$$

where $I_{Na}$ is the level of sodium current, $F$ is Faraday's constant, $Vol$ is the volume of one compartment of the model axon and $\alpha$ is a scale factor. According to this equation, at steady state, $I_{pump}$ is equal to one third of $I_{Na}$. Because of their comparatively small magnitude, the contributions of $I_{Leak}$ and $I_h$ to $[Na^+]_{in}$ were ignored and changes in $[Na^+]_{in}$ were modeled to depend solely on $I_{Na}$. The equilibrium potential of Na$^+$ in the model axon was calculated according to the change in intracellular sodium concentration from the Nernst equation:

$$E_{Na} = 58 \log_{10} \frac{[Na^+]_{out}}{[Na^+]_{in}}.$$

In our model, we made the simplifying assumption that the values of $[Na^+]_{out}$ and the K$^+$ concentrations inside and outside the axon were not changed by activity.

We also tried other, more descriptive, models of the Na$^+$/K$^+$ pump (**Cressman et al., 2009**; **Yao et al., 2011**) in our simulations. These provided somewhat different dynamics but produced very similar qualitative and quantitative results, which for brevity are not shown.

### Axon stimulation

Spikes were generated in the axon by applying a stimulus current in the first compartment:

$$I_{app}^0(t) = \begin{cases} 1\,(\text{nA}), & t \in [t_i, t_i + dt] \\ 0, & \text{otherwise} \end{cases}$$

where $t_i$ and $dt$ (1 ms) are the stimulus time and duration, respectively. Spikes propagated along the length of the 1 cm axon and were recorded at two sites (0.3 and 0.7 cm; **Figure 1B**). The conduction delay was measured in NEURON using the time that the spike crossed the voltage threshold of $-40$ mV at the two recording sites. (Delays measured using action potential peak times produce practically identical results.) Delays produced along this 0.4 cm of axon were multiplied by a factor of 9.5 to mimic the conduction delays and velocities of the 4–5 cm biological axon shown in our

results for comparison. Different conduction velocities of consecutive spikes lead to gradually changing intervals over distance. Because axonal excitability changes between spikes depend on interval, conduction delays during repetitive spiking do not scale linearly with distance (*Kocsis et al., 1979*; *Bucher and Goaillard, 2011*; *Bucher, 2015*). However, at initial spike frequencies smaller than about 100 Hz, these nonlinearities are relatively subtle and only differentially affect delays over distances of tens of centimeters (*Moradmand and Goldfinger, 1995*; *Bucher and Goaillard, 2011*). We confirmed that the conduction delay for a longer model axon, measured at the same relative points, scaled reasonably linearly with the length of the axon.

## Stimulation patterns

We characterized the history-dependence of conduction delay with three different stimulus regimes that have been used before in the PD axon (*Ballo et al., 2012*). First, to describe delay as a function of time and stimulus frequency, we used 300 s *Poisson* stimulation protocols with different mean rates: 5, 10 and 19 Hz. The last value was chosen to match mean spike frequency in burst stimulation protocols (see below). It should be noted that the *Poisson* stimulation method is useful in producing a variety of inter-stimulus intervals, but this can be also achieved with other stimulation paradigms that incorporate such a variety, for example through rate adaptation.

Second, we used a paired-pulse protocol to obtain 'recovery cycle' measurements (*Krishnan et al., 2009*; *Bucher and Goaillard, 2011*). After a single conditioning stimulus, test stimuli were delivered at different intervals (between 10 ms and 10 s). The change of delay or conduction velocity was then analyzed as a function of stimulus interval.

Third, we used a burst stimulation protocol that mimicked normal ongoing rhythmic firing in the PD neuron (*Ballo and Bucher, 2009*; *Ballo et al., 2012*). It consisted of 300 bursts at 1 Hz burst frequency, and each burst consisted of 19 spikes (360 ms burst duration) in a parabolic instantaneous frequency structure, increasing from 32 Hz to 63 Hz and decreasing back to 32 Hz.

All simulations were run beginning with a 100 s interval of no stimulation to remove any transient and initial condition effects.

## Equations for predicting conduction velocity

Two equations have been shown to provide good approximations for conduction velocity of a single spike in the squid giant axon and Hodgkin-Huxley model axon. We tested if these equations yielded good estimates of conduction velocities during repetitive spiking. The first is derived based on boundary matching principles which we will refer to as the Matsumoto-Tasaki equation (*Matsumoto and Tasaki, 1977*; *Tasaki and Matsumoto, 2002*; *Tasaki, 2004*):

$$v = \sqrt{\frac{d}{8R_{total}R_iC_m^2}}\sqrt{\frac{(1-\kappa)^2}{1+\kappa}}.$$

In this equation, $v$ is the conduction velocity, $d$ is the diameter of the axon, $R_{total}$ is the total resistance of the membrane of unit area in the excited state, $R_i$ is the axial resistivity of the axon interior, $C_m$ is the membrane capacitance per unit area and $\kappa$ is defined as the ratio of $R_{total}$ to $R_{rest}$, which is the resistance of the membrane of unit area at rest. In our evaluations, as in those done by Matsumoto and Tasaki, the value of $\kappa$ was consistently close to 0 and did not make any demonstrable difference in the estimations. We therefore followed Matsumoto and Tasaki's approach and used the simplified equation

$$v = \sqrt{\frac{d}{8R_{total}R_iC_m^2}}.$$

The second equation is an explicit analytical expression developed for calculating conduction velocity in the Hodgkin-Huxley model (*Muratov, 2000*), which we will refer to here as the Muratov equation:

$$v = \frac{2}{3}\left(\frac{r^4\bar{\alpha}_m^3\bar{g}_{Na}h_0}{16R_i^4C_m^5}\right)^{1/8}$$

where $R_i$ and $C_m$ are as above, $r$ is the radius of the axon, $\bar{g}_{Na}$ is the maximum fast sodium conductance, $h_0$ is the value of the inactivation variable $h$ of $g_{Na}$ at the rest state and

$$\bar{\alpha}_m = \alpha_m(E_{Na}) - \alpha_m(V_{rest}), \alpha_m = \frac{m_\infty}{\tau_m^{Na}}$$

where $E_{Na}$ is the sodium equilibrium potential and $V_{rest}$ is the resting membrane potential. To use this equation for repetitive spiking, we used the trough voltage before each spike (the local minimum voltage immediately before the spike) as a proxy for the resting membrane potential.

## Sensitivity analysis

The sensitivity of conduction delay in the model to parameters was measured by examining how the different attributes that describe the slow- (STS) and fast-timescale (FTS) effects depend on small changes in the parameter values. For the STS effect, these attributes were the mean conduction delay and its coefficient of variation, measured for the final 20 s interval of simulation. For the FTS effect, we used quadratic fits to determine the minimum delay and the frequency at which it occurred, and the curvature of the quadratic fit at this point. To measure sensitivity, each parameter in the (reference) model was decreased or increased by 5% and 10% of its original value while other parameters were kept unchanged. It should be noted that $E_{Na}$ is a dynamic parameter, which is determined by the intracellular and extracellular concentration of Na$^+$. Therefore, the scaling factors were applied to the Nernst equation for Na$^+$ in order to rescale $E_{Na}$.

The 5% and 10% changes in each direction produced four new models for each parameter. Each new model was subjected to the same *Poisson* stimulation as the original model and the attributes of the fast and slow timescale effects were measured. We then normalized these measures by their values in the reference model. The resulting four data points, together with the reference value (at the origin) were fit with a line. The sensitivity of an attribute to the parameter was defined as the slope of the linear fit. For example, a sensitivity value of 1 meant that a $\pm$ 5% and $\pm$10% change in the parameter resulted in a $\pm$ 5% or $\pm$10% change in the attribute.

## Experimental data

For comparison of model results with experimental results, as well as for testing the empirical method of predicting conduction delay, we used experimental delay data from the PD axon obtained in our previous study (*Ballo et al., 2012*).

## Optimization

All optimization was performed using Powell's conjugate direction method (Powell's method; *Powell, 1994*), implemented in Matlab.

## Acknowledgements

Supported by NIH grants NS083319, MH60605 and NS058825.

## Additional information

### Funding

| Funder | Grant reference number | Author |
| --- | --- | --- |
| National Institute of Neurological Disorders and Stroke | NS083319 | Dirk Bucher Farzan Nadim |
| National Institute of Mental Health | MH060505 | Farzan Nadim |
| National Institute of Neurological Disorders and Stroke | NS058825 | Dirk Bucher |

The funders had no role in study design, data collection and interpretation, or the decision to submit the work for publication.

## Author contributions
YZ, Formal analysis, Validation, Investigation, Visualization, Methodology, Writing—original draft; DB, Funding acquisition, Methodology, Writing—review and editing; FN, Conceptualization, Supervision, Funding acquisition, Validation, Methodology, Writing—original draft, Project administration, Writing—review and editing

## Author ORCIDs
Dirk Bucher, http://orcid.org/0000-0003-4144-2895
Farzan Nadim, http://orcid.org/0000-0003-4144-9042

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
