## [Decision Letter]

Thank you for submitting your article "Ionic Mechanisms Underlying History-Dependence of Conduction Delay in an Unmyelinated Axon" for consideration by *eLife*. Your article has been favorably evaluated by Eve Marder (Senior Editor) and three reviewers, one of whom, Frances K Skinner (Reviewer #1), is a member of our Board of Reviewing Editors. The following individual involved in review of your submission has agreed to reveal their identity: Jeremy Niven (Reviewer #2).

The reviewers have discussed the reviews with one another and the Reviewing Editor has drafted this decision to help you prepare a revised submission.

Summary:

In this manuscript the authors examine action potential biophysics and use computational modelling to assess the history-dependence of conduction delay, as motivated by an unmyelinated invertebrate axon. The manuscript is well-written and well-organized with rationale, logic and motivation clearly presented. The reviewers felt that it should be of interest to neuroscientists from a variety of backgrounds, including those interested in the biophysics and those interested in neural coding and neuropathies. They use parameter screening and NEURON simulations to present findings that can explain the slow time scale via *I_h_* and Na-K pump dynamics along with an empirical equation that can be obtained by a nonlinear combination of peak and threshold potentials.

Essential revisions:

While the manuscript itself was clear, some concerns regarding resting and reversal potentials were raised as well as the need for better explanations in some places. Specifically:

1) The authors mention that they have fixed the reversal potential of K^+^ in the axon model to -70 mV to prevent the occurrence of an after-hyperpolarisation. Have they checked whether this will affect any of their conclusions? We appreciate that they state that the after-hyperpolarisation is missing from axonal recordings of APs but this could be due to leaky recordings, or recordings being made in passive regions of an otherwise active axon (as occurs just prior to the output synapses in some arthropod neurons). It may be worth checking that the results are robust to a shift of the K^+^ reversal potential to something more conventional for an arthropod neuron such as -80 or -85 mV.

2) Near the beginning of the Results section the authors spend some time discussing whether the slow effects they observe could be reproduced by a K^+^ conductance rather than the Na+/K^+^ ATPase. I was a little concerned that this might appear to be a bit of a straw-man argument. The two will not be equivalent because of the time constants governing their dynamics, and because only in one case is the current accompanied by a conductance change. Surely, the choice of sufficiently long time constants would allow a voltage-gated conductance to replace the ATPase as long as the reversal potential was correct?

Could this be further examined as a novel mechanistic explanation to consider that the observation that the pump state is a more potent determinant of velocity than a similarly slow K conductance? Perhaps this could lead to an experimentally testable distinction of the two potential mechanisms. The present treatment in the manuscript is restricted to effects in line with the previous experimental work. This is not required for the present revision, but the authors may wish to consider this.

3) The origin of the resting and reversal potential/s in the model was a bit confusing. The K^+^ reversal potential is set to -70 mV and the *I_h_* reversal potential presumably represents a mixed conductance of K^+^ and Na^+^ ions, but what happens to change this mixed conductance (and the reversal potential) in the presence of dopamine? Is the leak set to -65 mV to match the observed resting potential and, if so, is there experimental evidence to support such a depolarising leak in these neurons?

4) The sensitivity analysis description could be expanded to be more clear since it is a bit confusing as it stands given the earlier results/statements. For example, STS is mediated by *I_pump_*, but they then show that *g_h_* has a strong effect, whereas the sensitivity analysis shows other parameters have important contributions. Similarly, FTS is not only affected by *I_h_* but also by *I_pump_* and much by EK and the sensitivity analysis shows only a comparatively weak effect of *g_h_*. Instead of the conclusions reached in the text, it looks like it is either a complex problem with many covariates, or the simple determinant has not been correctly identified. Presumably this is because one is looking at parameter sensitivity and not sensitivity to *I_h_* itself, as stated toward the end of the section, but would perhaps be more helpful if this is stated more up front along with an expanded description to avoid confusion.

5) The authors are encouraged to expand their empirical equation discussion to have a wider appeal, perhaps in terms of a strategy. That is, if new currents were added/considered in the system and the consideration of time scale, how would one proceed?

---

## [Author Response]

*Essential revisions:*

*While the manuscript itself was clear, some concerns regarding resting and reversal potentials were raised as well as the need for better explanations in some places. Specifically:*

*1) The authors mention that they have fixed the reversal potential of K^+^ in the axon model to -70 mV to prevent the occurrence of an after-hyperpolarisation. Have they checked whether this will affect any of their conclusions? We appreciate that they state that the after-hyperpolarisation is missing from axonal recordings of APs but this could be due to leaky recordings, or recordings being made in passive regions of an otherwise active axon (as occurs just prior to the output synapses in some arthropod neurons). It may be worth checking that the results are robust to a shift of the K^+^ reversal potential to something more conventional for an arthropod neuron such as -80 or -85 mV.*

Intracellular recordings of the PD axons (and other axons in the same nerve) were performed in an active and propagating region, centimeters from the terminals (Ballo and Bucher, 2009; Ballo et al., 2012). None of these recordings show after-hyperpolarizations unless depolarized to -60 mV or above. We are currently recording the K reversal potential in axons and our data is consistent with the model and predicts that EK is significantly more depolarized in axons compared to the somata. This has been previously reported for the squid giant axon by John Clay (1998) in J. Neurphys, as quoted from the Abstract:

“accumulation/depeletion [sic] of K^+^ in the space between the axon and the glial cells surrounding the axon, which is significant even during a single action potential and which can account for the 15-20 mV difference between the potassium equilibrium potential EK and the maximum afterhyperpolarization of the action potential.”

We discuss Clay’s point (subsection “The dependence of conduction delay on slow activity-dependent hyperpolarization”, fourth paragraph). Nonetheless, as the reviewer suggests, we repeated the simulations of our main result (Figure 2) with EK=-80 mV (and corresponding adjustments to the INa activation and inactivation parameters) and found that although there are slight quantitative changes, the qualitative results remain valid.

*2) Near the beginning of the Results section the authors spend some time discussing whether the slow effects they observe could be reproduced by a K^+^ conductance rather than the Na+/K^+^ ATPase. I was a little concerned that this might appear to be a bit of a straw-man argument. The two will not be equivalent because of the time constants governing their dynamics, and because only in one case is the current accompanied by a conductance change. Surely, the choice of sufficiently long time constants would allow a voltage-gated conductance to replace the ATPase as long as the reversal potential was correct?*

We agree with the reviewer that a slow ionic current changes the input conductance whereas a pump current shouldn’t. Our main argument here was that the driving force for the voltage-gated current is very small in between spikes, and we assumed that this would be true even if *E_K_* was 10 or 15 mV more hyperpolarized. However, as the reviewer suggests, if the slow K^+^ current has a reversal potential that is much lower than the resting potential, it is in fact possible to change both the peak and trough voltages of the action potentials and, in a manner, have this current to replace the pump current for the purpose of changing AP delays. This point is by no means central to our argument so we modified the Discussion accordingly and added the point that the reviewer suggests. (subsection “A slow K^+^ current fails to produce the STS effect”, last paragraph and subsection “The dependence of conduction delay on slow activity-dependent hyperpolarization”, third paragraph).

*Could this be further examined as a novel mechanistic explanation to consider that the observation that the pump state is a more potent determinant of velocity than a similarly slow K conductance? Perhaps this could lead to an experimentally testable distinction of the two potential mechanisms. The present treatment in the manuscript is restricted to effects in line with the previous experimental work. This is not required for the present revision, but the authors may wish to consider this.*

The reviewer is correct that such a comparison in the model could potentially produce an experimental design in support of the pump current determining the STS effect. It is indeed an interesting point that a pump current and a conductance do different things to spike conduction. The pump current only affects the membrane potential, while a conductance, even one with no appreciable effect on the membrane potential, may affect conduction because it adds to the total input conductance and reduces the effect other currents have on the membrane potential.

We believe, however, that this message is not central to our current point and that a more substantial comparison is required to rule out the role of a very slow K^+^ current in shaping conduction delays. We therefore toned down our argument in ruling out the role of such a current, as stated in the previous point.

*3) The origin of the resting and reversal potential/s in the model was a bit confusing. The K^+^ reversal potential is set to -70 mV and the I_h_ reversal potential presumably represents a mixed conductance of K^+^ and Na^+^ ions, but what happens to change this mixed conductance (and the reversal potential) in the presence of dopamine? Is the leak set to -65 mV to match the observed resting potential and, if so, is there experimental evidence to support such a depolarising leak in these neurons?*

While *I_h_* is a mixed conductance, usually based on channels with a higher permeability to K^+^, the difference in driving forces in the voltage-range it is activated in makes it essentially just a Na^+^ current. Either way, dopamine may have a small (and unexplained) effect on the reversal potential of this current in the PD axon (Ballo et al. 2010), but the main effect is a conductance increase and shift in activation curve that results in a 2-3 fold increase in conductance in the voltage range of the baseline membrane potential. As the contribution of K^+^ to this current is minimal, it should not be very sensitive to the exact value of *E_K_*. The leak is set to the resting membrane potential, and -65 mV is not depolarized for *E_leak_* in lobster neurons and is consistent with the recordings of the PD axon (Ballo & Bucher 2009).

*4) The sensitivity analysis description could be expanded to be more clear since it is a bit confusing as it stands given the earlier results/statements. For example, STS is mediated by I_pump_, but they then show that g_h_ has a strong effect, whereas the sensitivity analysis shows other parameters have important contributions. Similarly, FTS is not only affected by I_h_ but also by I_pump_ and much by EK and the sensitivity analysis shows only a comparatively weak effect of g_h_. Instead of the conclusions reached in the text, it looks like it is either a complex problem with many covariates, or the simple determinant has not been correctly identified. Presumably this is because one is looking at parameter sensitivity and not sensitivity to I_h_ itself, as stated toward the end of the section, but would perhaps be more helpful if this is stated more up front along with an expanded description to avoid confusion.*

The reviewer is correct about the contributions of *I_pump_* and *I_h_*. We tried to clarify the apparent conflict between the influence of these currents and the weak effect of *g_h_* in the sensitivity analysis in the following sections of the Results and Discussion:

“Note that a small sensitivity value does not imply that the model attribute is not dependent on the parameter, but rather that small changes in the parameter do not affect that attribute strongly or monotonically. The large effect of *I_pump_* on conduction delay and history-dependence seen in Figure 3 was because of several-fold changes in pump activation over several minutes. In contrast, the comparatively small ( ± 5% and ± 10%) changes of *I_pump_* used here had little effect on sensitivity values (not shown).”

“It should be noted that the small level of sensitivity to the various currents is because sensitivity analysis only examines small changes in the parameters of these currents which would not greatly alter the membrane potential. However, some of these currents, as shown for the pump current and *I_h_*, undergo large changes that have substantial effects on delay.”

*5) The authors are encouraged to expand their empirical equation discussion to have a wider appeal, perhaps in terms of a strategy. That is, if new currents were added/considered in the system and the consideration of time scale, how would one proceed?*

We expanded the discussion of this equation (subsection “The relevance of the findings to other axons”).